# Caloric restriction delays age-related methylation drift

Shinji Maegawa[1,5], Yue Lu[2], Tomomitsu Tahara[1], Justin T. Lee[1], Jozef Madzo[1], Shoudan Liang[3], Jaroslav Jelinek[1], Ricki J. Colman[4] & Jean-Pierre J. Issa[1]

In mammals, caloric restriction consistently results in extended lifespan. Epigenetic information encoded by DNA methylation is tightly regulated, but shows a striking drift associated with age that includes both gains and losses of DNA methylation at various sites. Here, we report that epigenetic drift is conserved across species and the rate of drift correlates with lifespan when comparing mice, rhesus monkeys, and humans. Twenty-two to 30-year-old rhesus monkeys exposed to 30% caloric restriction since 7–14 years of age showed attenuation of age-related methylation drift compared to ad libitum-fed controls such that their blood methylation age appeared 7 years younger than their chronologic age. Even more pronounced effects were seen in 2.7–3.2-year-old mice exposed to 40% caloric restriction starting at 0.3 years of age. The effects of caloric restriction on DNA methylation were detectable across different tissues and correlated with gene expression. We propose that epigenetic drift is a determinant of lifespan in mammals.

---

[1] Fels Institute for Cancer Research & Molecular Biology, Lewis Katz School of Medicine at Temple University, Philadelphia, Pennsylvania 19140, USA. [2] Department of Epigenetics and Molecular Carcinogenesis, The University of Texas MD Anderson Cancer Center, Houston, Texas 77030, USA. [3] Department of Biostatistics and Computational Biology, The University of Texas MD Anderson Cancer Center, Houston, Texas 77030, USA. [4] Wisconsin National Primate Research Center and Department of Cell and Regenerative Biology, University of Wisconsin, Madison, Wisconsin 53715, USA. [5] Present address: Department of Pediatrics, The University of Texas MD Anderson Cancer Center, Houston, Texas 77030, USA. Correspondence and requests for materials should be addressed to S.M. (email: smaegawa@mdanderson.org)

The only intervention known to lengthen lifespan in taxonomically diverse organisms is caloric restriction (CR), a reduction in food intake without malnutrition. Evidence that mammalian longevity could be increased first emerged in 1935 in a rat study showing that CR-extended lifespan[1]. CR prolongs lifespan in most mouse strains examined[2]. This phenomenon has been extended to primates in a long-term experiment showing increased survival and reduction of age-related diseases including diabetes, cancer, cardiovascular disease, and brain atrophy in CR monkeys (rhesus macaques)[3]. Although health benefits and disease prevention have clearly been observed, the molecular basis for the delayed aging remains unknown.

During normal aging, gene expression and epigenetic modification changes occur in a tissue-specific manner. In mammals, DNA methylation occurs almost exclusively within the context of CpG dinucleotides and an estimated 80% of all CpG sites are methylated[4]. CpG islands (CGIs) are clusters of CpG dinucleotides that are often located around gene transcription start sites (TSS)[5]. Although most CGIs are unmethylated in normal human tissues, methylation changes of a small subset of genes can be seen in normal healthy individuals in aging tissues. Several groups identified age-related methylated (ARM) genes in human whole blood[6–9], and reported that this methylation could be used as a biomarker to predict biological age (epigenetic age)[9–11].

CGI methylation has also been suggested to be a good biomarker for the progression of cancers and diabetes[12, 13].

A number of tumor suppressor genes are silenced by promoter CGI methylation in cancers[14]. In parallel, genome-wide DNA hypomethylation is thought to play an important role in genomic instability and carcinogenesis[15]. Because cancer is largely a disease of aging, we and others proposed that age-related epigenetic changes initiate tumorigenesis[16–18]. Indeed, age-related DNA methylation drift is accelerated in age-related diseases including cancers, diabetes, and chronic inflammation[19–24].

Here, we studied CR as an intervention that could potentially influence age-related DNA methylation drift, and compared methylation status by genome-wide DNA methylation profiling among mouse (*Mus musculus*), rhesus monkey (*Macaca mulatta*), and human (*Homo sapiens*) blood cells. We find that the rate of epigenetic drift correlates with lifespan and that CR protects against DNA methylation deregulation with age.

## Results

**Age-related methylation drift is conserved across species**. DNA methylation drifts with age in mice and humans[25–28] but comparable analysis with lifespan is lacking. The maximum longevity of mice, rhesus monkeys, and humans is 4, 40, and 122.5 years, respectively (The Animal Ageing and Longevity Database)[29–31]. To assess conservation in methylation drift among species, we utilized a quantitative deep sequencing-based method Digital Restriction Enzyme Analysis of Methylation (DREAM)[32, 33] for DNA methylation analysis of whole blood samples from 19 mice

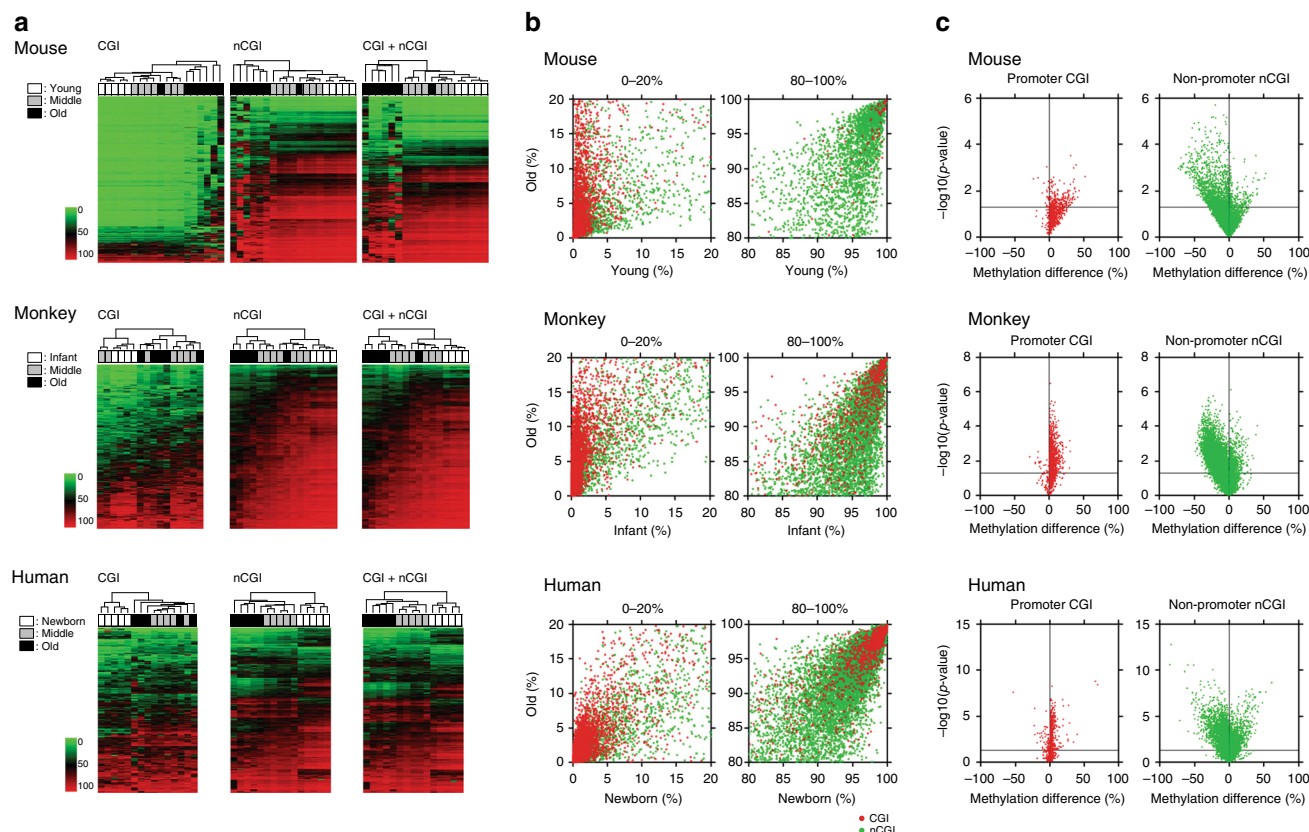

**Fig. 1** DREAM DNA methylation analysis of whole blood DNA from different species. **a** Unsupervised hierarchical clustering analysis of CpG sites in CpG island (CGI), non-CGI, and all genomic regions. The *green to red color scale* indicates the methylation percentage. The color codes for age are shown on the *left*. **b** DNA methylation in old vs. newborn/infant/young. Average DNA methylation level of each CpG site in old individuals (*y* axis) is plotted against that in newborn/infant/young individuals (*x* axis). The *red* and *green dots* represent CGI and non-CGI CpG sites, respectively. The low range (0–20%) and high range (80–100%) of methylation status are shown. **c** Volcano plots show CpG sites differentially methylated between old and newborn/infant/young. Plots on *left* are sites in promoter CGI (*red*) and on *right site* are sites in non-promoter non-CGI (*green*). The promoter region is defined as −1 kb ≤ TSS ≤ + 500 bp. The methylation difference between old and newborn/infant/young is shown on the *x* axis, the *p*-value (in −log10 scale) on the *y* axis. The *horizontal line* indicates *p*-value at 0.05

### Hypermethylated genes

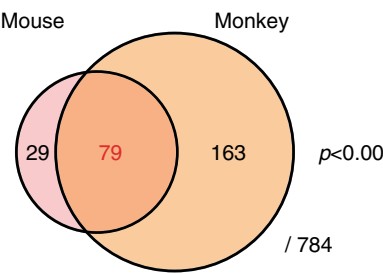

Mouse Monkey

29 79 163 $p < 0.001$

/ 784

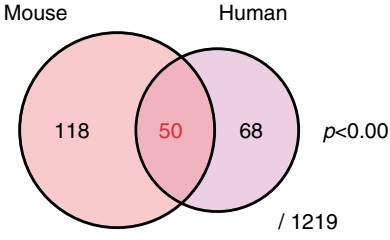

Mouse Human

118 50 68 $p < 0.001$

/ 1219

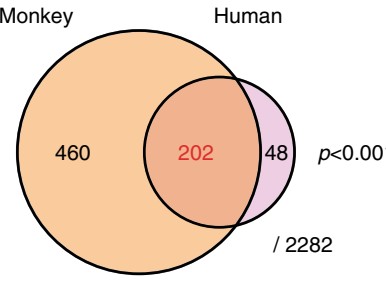

Monkey Human

460 202 48 $p < 0.001$

/ 2282

**Fig. 2** Methylation analysis by DREAM in mouse, monkey, and human DNA. Area-proportional Venn diagrams of overlapping gene promoters (−1kb < TSS <+ 500 bp) showing age-related methylation drift in whole blood in each combination. The denominators represent the number of homologous genes with high-quality sequence data (sequencing depth ≥ 100 reads in 75% of cases), which are detectable between two species in each comparison. We counted the number of genes using human homolog gene names of mouse and monkey genes in each comparison. A $\chi^2$-test using 2×2 tables (Supplementary Table 12) was used to calculate *p*-values for the significance of the overlaps. *p*-values are indicated on the *right side*

(age; 0.3–2.8 years), 16 monkeys (age; 0.8–30 years), and 16 humans (age; 0–86 years) (Supplementary Tables 1, 2). We used cord blood to represent age zero in humans. Cord blood samples have been used previously for DNA methylation studies in aging[6], [34–37]. DNA methylation profiles in cord blood cell specimens can be potentially affected by the presence of nucleated red blood cells in case of a low gestational age[38, 39]. This was likely not the case in our study because all cord blood samples we tested showed a very high concordance of methylation values (Pearson $r > 0.95$) (Supplementary Table 3).

Unsupervised hierarchical clustering analysis of CpG sites with high sequencing depth (≥100 reads) and variable methylation (standard deviation >10%, representing 20.8, 18.7, and 5.5% of sites in mice, monkeys, and humans, respectively) revealed clear clustering by age in all three species (Fig. 1a). Age-related gains of methylation were most pronounced in CGI sites that were unmethylated (<5%) in young individuals (Fig. 1b); out of these, the highly variable sites increased from a mean (±standard error of the mean; SEM) of $2 \pm 0.1\%$ in the young to $18 \pm 5\%$ in the old mice ($p = 0.03$, unpaired *t*-test with Welch's correction).

Corresponding numbers were $2 \pm 0.3\%$–$22 \pm 3\%$ in young vs. old monkeys ($p = 0.002$) and $3 \pm 0.5\%$–$20 \pm 4\%$ in newborn vs. old humans ($p = 0.009$). Conversely, age-related hypomethylation occurred at highly methylated (>90%) non-CGI sites (Fig. 1b). The variable of these decreased from an average methylation of $94 \pm 0.4\%$ in the young to $78 \pm 4\%$ in the old mice ($p = 0.003$). Corresponding numbers were $94 \pm 0.3\%$–$73 \pm 4\%$ in infant vs. old monkeys ($p = 0.007$) and $93 \pm 1\%$–$74 \pm 2\%$ in newborn vs. old humans ($p < 0.001$). The sample sizes for all comparisons were sufficient to give statistical power >0.8 (Supplementary Table 4). These differences are also evident in volcano plots for promoter CGI sites and non-promoter non-CGI sites (Fig. 1c). Generally similar results were seen for all three species, but with more variability in humans possibly due to a higher degree of heterogeneity (genetic, diet etc.).

To identify ARM sites more precisely, we computed Spearman's rank correlation ($r$) between methylation and age for each CpG site and assigned an empirical *p*-value for each $r$ based on a data set of 1000 random permutations of ages. Based on the distributions of observed and permuted correlation coefficients (Supplementary Fig. 1; Supplementary Table 5), CpG sites that showed $r \geq 0.5$ (hypermethylation) or $r \leq −0.5$ (hypomethylation), empirical $p < 0.05$, and average methylation ≥1% were selected for further analyses. Ingenuity pathway analysis of hypermethylated genes showed enrichment for developmental processes, gene networks involved in cancer and cardiovascular disease, and molecular and cellular functions including cell development, signaling, growth and maintenance (Supplementary Tables 6–8). There was general conservation in the pathways that affected across species.

To assess the effects of blood composition on age-related methylation status detected in whole blood, we performed DREAM methylation analysis using purified subpopulations of blood cells: granulocytes ($n = 6$), CD34+ cells ($n = 2$), and T-cells ($n = 3$) and compared these to whole blood samples ($n = 16$). We detected 222, 1045, and 1923 sites significantly hyper- or hypomethylated (methylation differences ≥2%, sequence depth ≥100 reads in each site, false discovery rate (FDR) < 0.05) in granulocytes, CD34+ cells, and T-cells, respectively, compared to the whole blood. A limited overlap of 0.2–10% between these cell-type-specific sites and the ARM sites determined using whole blood suggested that age-related methylation drift cannot be explained by variability in blood cell subtypes (Supplementary Fig. 2; Supplementary Table 9). This is in agreement with a previous study using neutrophils, eosinophils, monocytes, and lymphocytes[6]. Moreover, age-related methylation affects both CGIs and non-CGIs, while tissue-specific methylation affects predominantly non-CGI sites[40]. Thus, it is likely that the changes observed here are independent of tissue composition.

We also compared ARM genes obtained by DREAM to those obtained by other methods and reported in previous studies[7–9] (Supplementary Tables 10, 11). Overall, there was a significant overlap, but DREAM detected a much higher number of drifted genes likely reflecting different methods and the higher quantitation ability of DREAM.

To directly compare methylation drift across the three species, we focused on promoter regions (Supplementary Data 1–3) because other sites (e.g., intergenic CGI sites) show little sequence conservation. We analyzed 2282 genes that had high sequencing coverage (minimum 100 reads in 75% of cases) in both human and monkey blood. 250 genes (11%) showed hypermethylation in human DNA and 202 of those (81%) were also hypermethylated in monkey DNA ($p < 0.001$, $\chi^2$-test). In a similar comparison, 73% (79/108) of the homologs of mouse hypermethylated ARM genes also showed age-related hypermethylation in monkeys ($p < 0.001$, $\chi^2$-test). Finally, 42% (50/118) of human

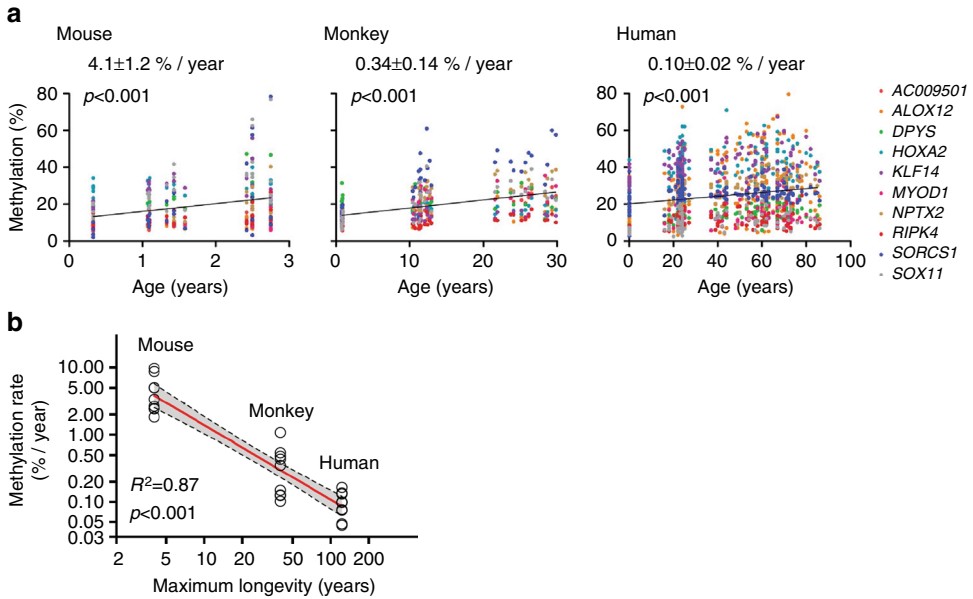

**Fig. 3** Cross-species comparison of aging drift. **a** The association between age (x axis) and methylation percentage of 10 hypermethylated ARM genes (y axis) homologous in each species. *Dots* represent methylation values at 10 genes in each individual (mouse; young, $n = 6$, middle, $n = 13$, old, $n = 12$, monkey; infant, $n = 12$, middle, $n = 15$, old, $n = 12$, human; newborn, $n = 13$, young, $n = 54$, middle, $n = 27$, old, $n = 45$). p-values of the linear regression slope were calculated by t-test (two-tailed). **b** The association between methylation drift rate and lifespan. The x axis represents maximum longevity of each species (mouse; 4 years, monkey; 40 years, human; 122.5 years), and the y axis represents methylation changes per year. The scales are logarithmic. The maximum longevities of the three species were obtained from The Animal Ageing and Longevity Database (http://genomics.senescence.info/species/). The R-squared values and their p-values (F-test, two-tailed) were calculated

hypermethylated ARM genes were also detected as hypermethylated ARM genes in mice ($p < 0.001$, $\chi^2$-test) (Fig. 2; Supplementary Table 12). These data suggest that aging methylation drift is evolutionarily conserved across species. Moreover, we are likely underestimating conservation because of the limited number of analyzed CpG sites and samples.

**Methylation drift correlates with gene expression changes**. To analyze the impact of age-related DNA methylation changes on gene expression, we first correlated our DREAM data with gene expression using published RNA-seq data set from human whole blood (GSE53655)[41]. We divided the 8663 unique genes with methylation data (35,379 CpG sites) into four groups based on the ranking of expression levels. Methylation in a window of −1 to +1 kb relative to TSS was $15 \pm 1\%$ (mean ± SEM) in unexpressed genes, $8 \pm 0.4\%$ in genes expressed at low levels, and $4 \pm 0.2\%$ in genes with moderate or high levels of expression (Supplementary Fig. 3). We next analyzed an RNA-seq data set of age-related gene expression in human monocytes (GSE60216)[42]. There were 328 genes that changed gene expression (fold change > 2, $p < 0.05$, DESeq[43]) for which promoter methylation was available, and there was a significant negative correlation between methylation drift and change in gene expression (Spearman $r = -0.20$, $p < 0.001$, two-tailed). Genes showing gains of expression with age had significant demethylation, and a fraction of genes that lost expression with age showed striking concomitant gains of DNA methylation (Supplementary Fig. 4).

**Age-related methylation correlates with lifespan**. To validate DREAM results with an orthogonal technology and to extend the number of samples, we used bisulfite pyrosequencing assays to study DNA from 31 mice (age; 0.3–2.8 y), 39 monkeys (age; 0.8–30 y), and 139 humans (age; 0–86 y). We selected genes based on age-related drift detected in at least one species (Supplementary Table 13) and separately based on prior

publications[20, 26–28, 44]. Supplementary Figure 5 shows CpG maps of the genes analyzed, along with the location of the regions amplified. We studied 34 genes (24 showing hypermethylation and 10 showing hypomethylation) in mice; 29 showed statistically significant differences between methylation levels in old and young mice (Supplementary Fig. 6a; Supplementary Table 14). We also studied 36 genes in monkey DNA; 33 showed age-related drift (Supplementary Fig. 6b; Supplementary Table 15). In four individual monkeys, we analyzed three genes in DNA from peripheral blood mononuclear cells sampled at two-time points 4–5 years apart (Supplementary Table 13). Of 12 comparisons (three genes and four animals), nine cases showed age-related differences consistent among the animals ($p = 0.02$, binomial distribution). No methylation changes were detected in the remaining three cases (Supplementary Fig. 7). Finally, we studied 16 genes in human DNA and all genes showed significant drift with age (Supplementary Fig. 6c; Supplementary Table 16). Hypermethylation and hypomethylation occurring with aging could be seen as a regression to the mean. We previously reported age-related methylation changes (increased and decreased patterns in promoter regions) showing increased epigenetic noise (with increased variabilities in older populations) in multiple tissue types in mice[20, 44]. Hierarchical clustering analyses of the pyrosequencing data showed clear age-related patterns (Supplementary Fig. 8). Of note, some of the genes that showed significant changes by pyrosequencing (selected based on prior studies in cross-species comparisons) were not detected as changed by the DREAM assays (Supplementary Table 13), suggesting that the genome-wide study may have underestimated the extent of changes and conservation.

Having validated the DREAM data, we next compared age-related methylation drift by studying 10 genes that had a high level of sequence conservation and showed age-related hypermethylation in all three species (Supplementary Table 13). We employed a multilevel linear mixed effect model to calculate methylation drift at 10 homologous genes and obtained slopes

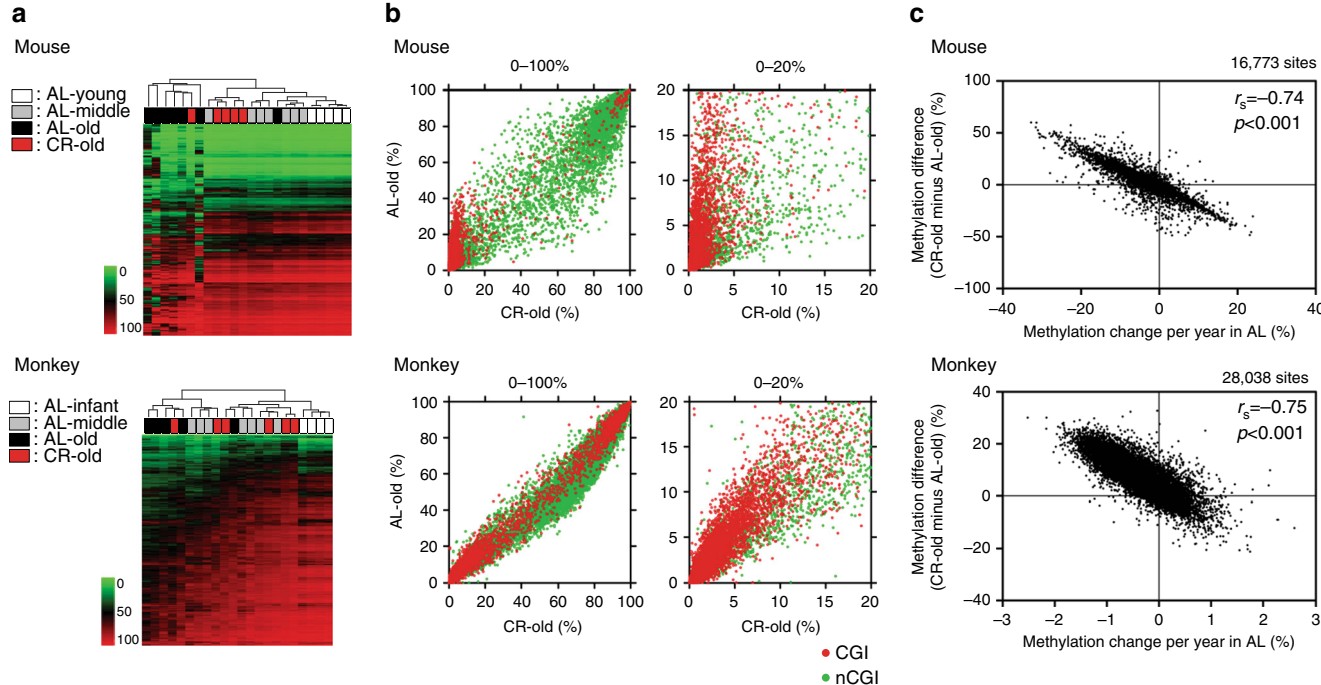

**Fig. 4** DREAM analysis in CR animals. **a** Unsupervised hierarchical clustering analysis of methylation values in all genomic regions. The *green* to *red scale* indicates the methylation percentage. The color codes for age and caloric status are shown on the *left*. **b** DNA methylation in AL old animals vs. CR old animals. Average DNA methylation level of each CpG site in CR old individuals (*x* axis), methylation in AL old individuals is shown on the *y* axis. The *red* and *green dots* represent CpG sites within CGI and non-CGI, respectively. The full range (0–100%) of methylation level is shown on the *left* and the low range (0–20%) is magnified and shown on the *right*. **c** Correlation between the effects of CR and age-related methylation drift in AL animals. The *x* axis shows methylation changes per year in AL-fed animals. Positive/negative value means methylation increase/decrease with age, respectively. The *y* axis shows the differences of methylation percentage between CR old and AL old animals. Each *dot* represents a CpG site. Spearman *r* values and the corresponding two-tailed *p*-values were calculated

representing methylation drift per year in each species (Supplementary Table 17). The drift rates (mean ± SEM) were 4.1 ± 1.2% per year in mice, 0.34 ± 0.14% per year in monkeys, and 0.10 ± 0.02% per year in humans (Fig. 3a). Methylation drift was thus inversely proportional to longevity (Fig. 3b). Similar results were obtained when considering all hypermethylated genes tested regardless of conservation (5.1 ± 0.4% per year in 24 genes in mice, 0.47 ± 0.02% per year in 24 genes in monkeys, 0.09 ± 0.01% per year in 14 genes in humans) (Supplementary Table 17). Thus, methylation drift correlated strongly with lifespan across these three mammalian species.

**Caloric restriction delays DNA methylation drift**. To study the effects of CR on methylation drift, we analyzed 2.7–3.2-year-old mice exposed to 40% CR starting at 0.3 years of age. We also studied rhesus macaques exposed to 30% CR starting in middle age (age; 7–14 y) and analyzed at 22–30 years of age (CR treatment period; 15–21 years). Unsupervised hierarchical clustering of DREAM data in mice showed that four out of five CR animals studied (median age; 2.8 y) clustered with young animals while the ad libitum (AL)-fed older mice clustered separately (Fig. 4a). Principal component analysis showed that CR old mice were close to the young and middle age animals, while AL old samples showed a clear separation (Supplementary Fig. 9). The CR effect was most pronounced at CGI sites that were unmethylated in young animals (Fig. 4b). To reveal the CR effect, we compared methylation differences in CR vs. AL animals to the rate of methylation drift with age in AL mice and found a strong negative correlation (Spearman *r* = −0.74, *p* < 0.001, two-tailed, Fig. 4c). To illustrate this, an analysis restricted to those CpG sites that drift heavily with age showed striking effects of CR in mice

(Supplementary Fig. 10). This strong correlation suggests that CR counteracts aging drift and does not create novel methylation patterns. Similar results were seen in CR-exposed monkeys (*n* = 6; median age; 26 y) where there was also a negative correlation between methylation drift with age and the effects of CR (Spearman *r* = −0.75, *p* < 0.001, two-tailed, Fig. 4; Supplementary Figs. 9, 10). However, the effects of CR in monkeys were less pronounced, suggesting that CR severity (30 vs. 40% in mice) and duration (2/3 of lifespan vs. almost entire lifespan in mice) influenced the resulting methylation patterns.

We investigated whether CR was related to methylation drift by a multiple linear regression of methylation on age with an interaction for CR allowed. Based on the *p*-values < 0.05 provided by the regression results for each CpG site, we defined genes where age-related drift was significantly alleviated by CR (Supplementary Table 18). As expected, almost every gene detected by this model (Supplementary Table 18) was also detected as undergoing age-related methylation as listed in Supplementary Data 1, 2. Most of the genes that showed a significant effect of CR as indicated by negative coefficients overlapped with hypermethylated ARM genes and vice versa. (Supplementary Fig. 11; Supplementary Table 18). These data suggest that CR may diminish or eliminate methylation changes with age.

We used bisulfite pyrosequencing assays described earlier to validate the CR effects on DNA methylation drift. In mice, we studied 12 CR animals (age; 2.7–3.2 y) together with 31 AL animals (age; 0.3–2.8 y). All of the 24 genes hypermethylated with age showed lower methylation levels in CR animals. The differences were statistically significant (*p* < 0.05, unpaired *t*-test with Welch's correction) in 15 genes (Supplementary Fig. 12a; Supplementary Table 14) and the average methylation of all 24

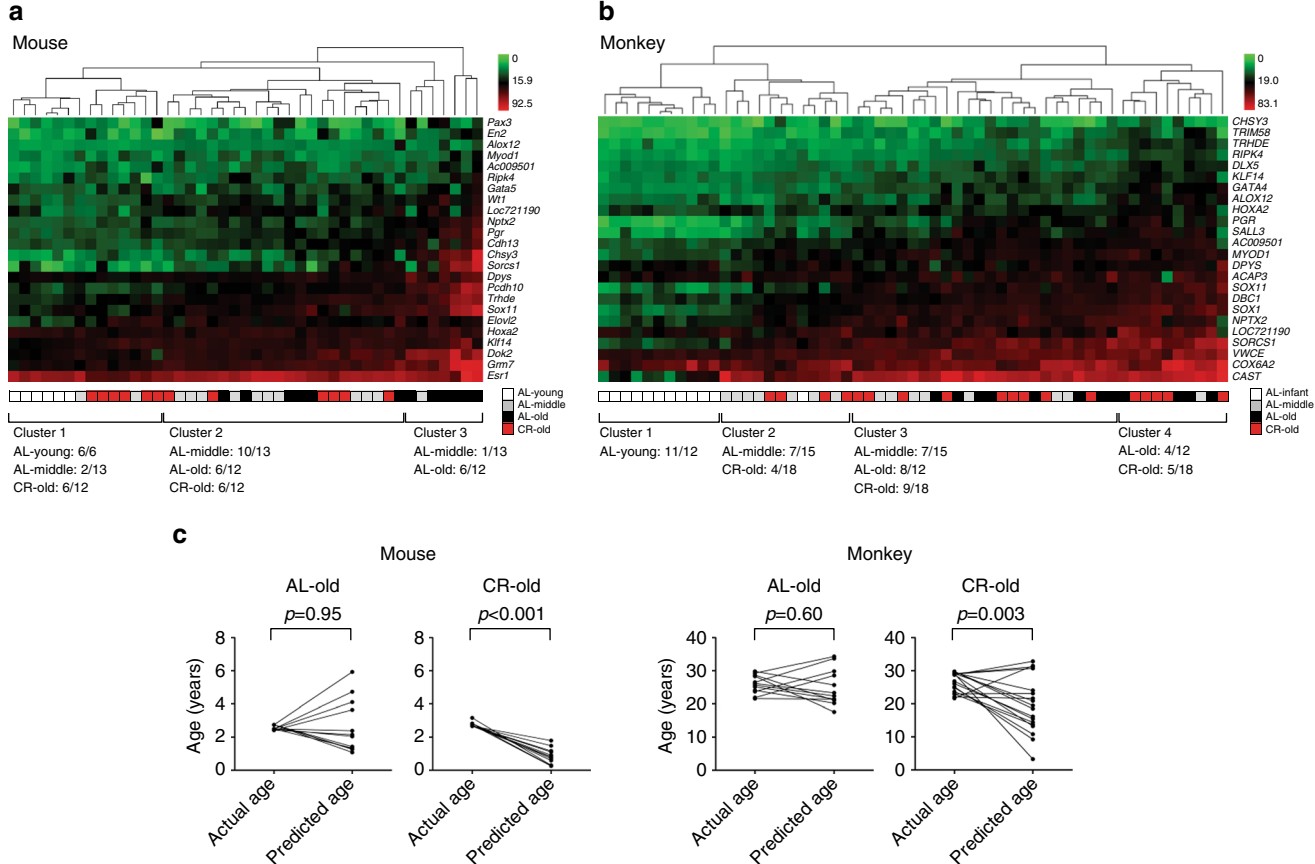

**Fig. 5** CR effects on aging drift detected by bisulfite pyrosequencing assays. **a**, **b** Unsupervised hierarchical clustering analysis of the loci assayed by bisulfite pyrosequencing showing aging methylation drift and the effect of CR in whole blood in mice and monkeys. The *green to red color scale* indicates the methylation percentage. The color codes of caloric status and age are shown on the *bottom-right*. **c** Differences between the age predicted based on methylation and the chronological age. Predicted age was calculated using the mixed linear model and data from bisulfite pyrosequencing assays. Each pair of *dots* connected by a *line* represents the difference between predicted age and chronological age in each individual. Paired *t*-test was used for *p*-value calculation

genes was 26 ± 2% in AL old mice and 17 ± 0.7% in CR old mice (*p* = 0.003, unpaired *t*-test with Welch's correction). We found a strong negative correlation (Spearman *r* = −0.92, *p* < 0.001, two-tailed) between the effects of CR (calculated as methylation in CR minus methylation in AL) and the rate of age-related methylation drift in AL animals (Supplementary Fig. 13). Hierarchical clustering analysis divided the samples into three clusters. Cluster 1 contained all AL young mice and half of CR old mice. Cluster 2 contained the rest of CR old mice while cluster 3 contained mostly AL old mice (Fig. 5a). We next analyzed 18 monkeys subjected to CR (age; 22–30 y) compared with the 39 AL animals (age; 0.8–30 y) described earlier. Lower-methylation levels were seen in CR old monkeys compared with AL old monkeys for most of the 24 genes analyzed (Supplementary Fig. 12b; Supplementary Table 15) and average methylation was 27 ± 0.7% in AL animals compared to 24 ± 0.9% in CR animals (*p* = 0.04, unpaired *t*-test with Welch's correction). As was the case for mice, there was a strong correlation between the effects of CR and age-related methylation drift (Spearman *r* = −0.78, *p* < 0.001, two-tailed) (Supplementary Fig. 13). Unsupervised hierarchical clustering of pyrosequencing results revealed four clusters; cluster 1 included mostly young animals, cluster 2 contained half of the middle aged plus 20% of the CR old animals, while clusters 3 and 4 had AL old animals and the rest of the CR old animals (Fig. 5b).

It has been previously shown that DNA methylation can be used as a predictor of chronological age[9–11]. To quantitate the CR effects on methylation in the process of aging, we used the

pyrosequencing data on 24 genes in mice and monkeys, and calculated a "methylation age" in CR animals based on the linear model built in AL animals (Fig. 5c). The 12 CR mice had an average chronologic age of 2.8 years and a methylation age of 0.8 years (*p* < 0.001, paired *t*-test). For 18 CR monkeys, the average chronologic age was 27 years while the predicted methylation age was 20 years (*p* = 0.003, paired *t*-test). Thus, in both mice and monkeys, CR was associated with a significantly lower-methylation age, though the effect was much more pronounced in mice (40% CR since early adulthood) than in monkeys (30% CR since middle age).

We next examined the tissue specificity of this process by studying DNA from spleen, bone marrow, liver, kidney, small intestine, and large intestine derived from the same mice we analyzed earlier. We tested 15 genes showing aging drift in the blood (12 hypermethylated genes and 3 hypomethylated genes) (Supplementary Table 13). Most of these genes showed age-related methylation drift in most of the tissues (Supplementary Table 19) with some exceptions. Kidney and liver generally showed less age-related hypermethylation at these loci, while large intestine showed even larger drift than blood. Blood, spleen, and kidney showed consistent hypomethylation drift, while liver, small, and large intestine had lower drift, and bone marrow showed hypomethylation in only 1/3 loci examined. Hierarchical clustering analyses showed broadly similar patterns in all tissues (Supplementary Fig. 14) with DNA from young animals clustering separately from old animals, while DNA from CR

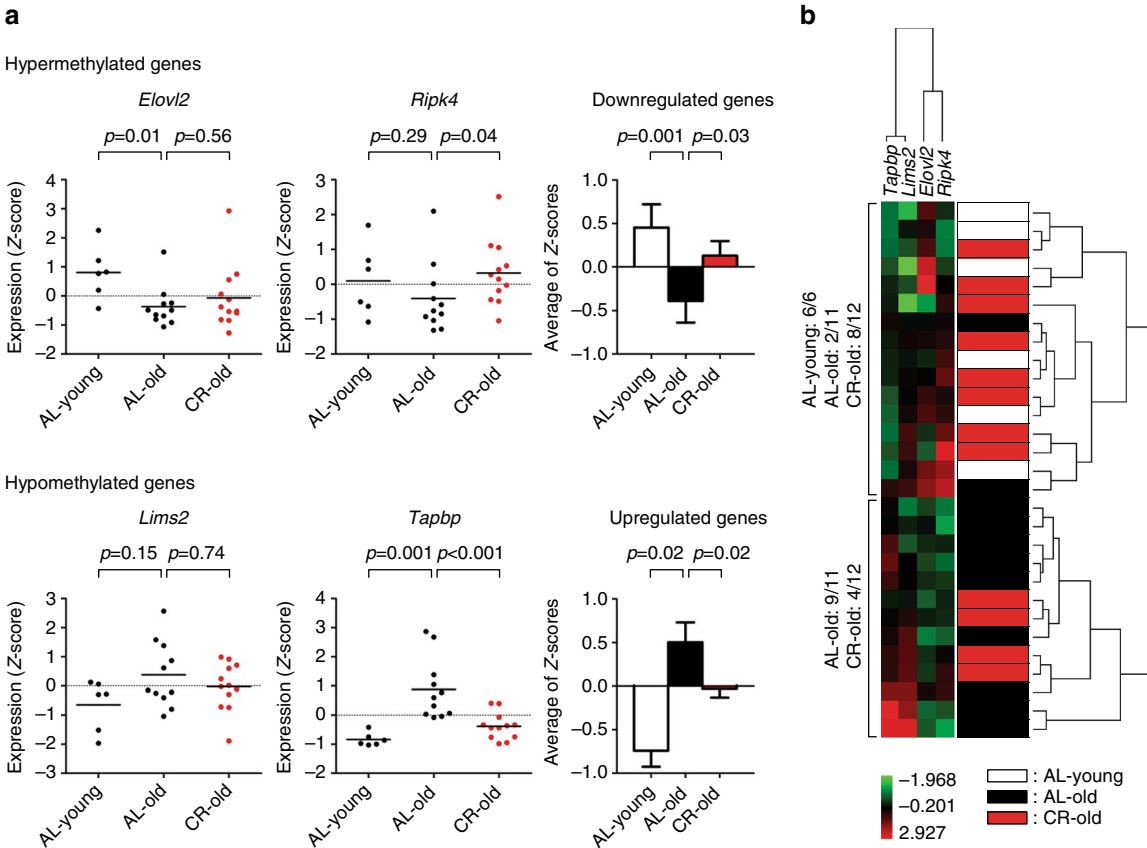

**Fig. 6** Expression of genes showing age-related methylation in mouse liver. **a** Gene expression in liver measured by quantitative RT-PCR. Values were normalized to the expression of *Gapdh*. The expression data for each gene were *Z*-score transformed. Each *dot* corresponds to one individual (AL young, $n = 6$, AL old, $n = 11$, CR old, $n = 12$). The values are presented as means of triplicate analyses. *Z*-scores were averaged for age-related hypermethylated and hypomethylated genes (bar graph, *right side*). *White, black,* and *red solid bars* represent the expression level of genes in AL young, AL old, and CR old samples, respectively. *Bars* represent standard errors. *p*-values were obtained using the unpaired *t*-test with Welch's correction. **b** Unsupervised hierarchical cluster analysis of gene expression data. Expression values were *Z*-score transformed. *Green boxes* indicate lower expression than the mean expression and *red boxes* depict higher expression. (*Red*; high expression, *green*; low expression.) The dendrogram is shown at the *right* of the figure. The color codes of caloric status and age are shown on the *bottom-right*

animals mostly clustered in-between young and old (for the most part). The patterns in blood, bone marrow, and spleen were similar, which likely reflects the fact that the genes were selected based on blood. Nevertheless, the same genes showed clear age-related clusters in small and large intestinal tissues as well, suggesting that the effects are not restricted to blood. These data also proved that age-related methylation changes occur in multiple tissues and the effect of tissue composition is limited. Similar to results in blood, we observed a strong negative correlation between the effects of CR on methylation and age-related methylation drift in most tissues (Supplementary Fig. 14). As before, we compared methylation age to chronological age in CR mice in all seven tissue types tested. Methylation age appeared lower in all tissues albeit with significant variability likely reflecting gene selection and tissue-specific variables. The differences between methylation age and chronological age were as follows (*p*-values; paired *t*-test): blood—1.7 years ($p < 0.001$); spleen—0.8 years ($p = 0.007$); bone marrow—0.8 years ($p < 0.001$); liver—1.5 years ($p = 0.02$); kidney—1.5 years ($p < 0.001$); small intestine—0.3 years ($p = 0.12$), and large intestine—0.5 years ($p < 0.001$) (Supplementary Fig. 14).

**CR effects correlate with gene expression**. To further test the impact of DNA methylation changes with age on gene expression,

we performed quantitative reverse transcription PCR (RT-PCR) analysis on RNA isolated from young and old mouse livers (young; $n = 6$, AL old; $n = 12$, CR old; $n = 12$). We selected four ARM genes for which we had pyrosequencing data in promoters (Supplementary Table 13) and which were expressed in normal liver based on publicly available data (*Elovl2*, *Ripk4*, *Lims2*, and *Tapbp*)[45]. All four genes showed expected aging trends (repressed by hypermethylation or activated by hypomethylation) and expression levels of CR animals were in between those of young and old animals (Fig. 6a). Hierarchical clustering based on expression levels divided the specimens into two distinct groups, one containing mainly young and CR old liver, with the other containing AL old liver (Fig. 6b).

Given the strong correlations between methylation drift, aging, and the effects of CR, we next compared this biological clock to telomere shortening, which had previously been proposed as a biological clock of aging[46, 47]. Using quantitative PCR to measure telomere length, we also found age-related telomere shortening in the blood of mice, monkeys, and humans (Supplementary Fig. 15), though the shortening was too small to be measured precisely. CR had no measurable effect on telomere length in either mice or monkeys (Supplementary Fig. 15) as previously reported in monkeys[48].

## Discussion

In this study, we found that DNA methylation drifts with age in an analogous manner in three mammalian species and that the rate of drift generally correlates with lifespan (Figs. 1, 3). Changes in DNA methylation occur during the aging process in mammals, and this age-related change in DNA methylation is accelerated in tumorigenesis. Methylation drift can be seen as an erosion of highly organized methylation patterns: focal unmethylated status of CGIs, distinctly different from the global methylated status of CpG sites in CpG poor regions[40]. Consistent with our previous studies[20, 44], we detected age-related methylation drift in multiple tissue types. One of the limitations of this study is the reliance on cross-sectional samples. While we did observe methylation changes in a limited number of paired samples obtained from the same individual in the span of 5 years, longitudinal prospective studies are needed to confirm our observations. Additionally, DNA methylation studies in pure cell populations would better address potential cell-specific effects on the methylation drift. We also found that CR, which prolongs lifespan in mice and monkeys, markedly delayed methylation drift and resulted in a significantly younger "methylation age" (Figs. 4, 5). Together with previous findings showing that chronic inflammation (which shortens lifespan) accelerates methylation drift[24], our data suggest that epigenetic drift is an excellent biomarker of lifespan. We also found that methylation drift generally correlates with gene expression changes (Supplementary Figs 3, 4; Fig. 6), suggesting it is a possible mediator of age-related functional decline and disease. We note that previous reports failed to see such a relationship with expression but the high accuracy of our method combined with the focus on promoters is likely to explain these differences[49–51].

Mechanisms underlying the vastly different lifespans across different species remain incompletely understood. Telomere shortening has been proposed as a potential mechanism regulating longevity but data across species have not been supportive of this hypothesis[52], and we did not find significant effects of CR on telomere length (Supplementary Fig. 15). While we analyze cells and tissues that are a mixture of stem cells and differentiated cells, it is likely that the most reproducible changes observed are happening in adult stem cells and are carried into differentiated somatic cells[16]. Changes that are restricted to differentiated cells would likely be lost upon cell death and contribute to increasing noise with age but not to the most dramatic and reproducible associations between methylation and age. The "stemness" origin of methylation drift in the hematopoietic system was experimentally established by prior studies[53], and differing stem cell turnover rates may explain the tissue specificity we observed—highest drift in large intestine for example (high stem cell turnover). While our data do not establish that methylation drift actually causes changes in lifespan, correlations with gene expression and prior data on methylation drift in stem cells suggest the possibility that restriction of stem cell plasticity by aging epigenetic drift may be a key regulator of lifespan. Stem cell dysfunction and exhaustion have recently emerged as a potential mechanism of age-related functional decline[54–56]. Given that epigenetic programs are largely set and reset during cellular replication[57], it is plausible that methylation drift reflects accumulated random epigenetic errors during stem cell division. The markedly differing rates of methylation drift across species would then suggest differing rates of stem cell turnover. Within a species, one can further speculate that inflammation increases methylation drift through cycles of injury/repair that stimulate stem cell turnover, while CR slows down drift by dampening stem cell turnover.

It has been known for decades that restricting the food intake of laboratory rodents extends their mean and maximum lifespan[1]. In monkeys, the WNPPRC study (from which we obtained samples for the current report) showed a clear effect of CR on aging[3]. However, another CR study using rhesus monkeys conducted at the National Institute on Aging (NIA) concluded that CR resulted in a lower cancer incidence but had no effect on lifespan[30]. The two studies differed substantially in the diets they used, possibly accounting for the different outcomes[58]. A comparison of epigenetic drift between the studies might help shed light on the differing outcomes. Data on CR and longevity are not available for humans. However, obesity increases the risk of chronic age-related diseases, such as type 2 diabetes, heart disease, osteoarthritis, and certain types of cancer (including colorectal, breast, pancreatic, and prostatic)[59, 60], and thus constitutes a major and rising global health problem. Our data in monkeys showed a positive correlation between methylation drift in whole blood and body mass index (Spearman $r = 0.38$, $p = 0.04$, two-tailed) (Supplementary Fig. 16). Recent data in humans also showed that body mass index is associated with DNA methylation in human whole blood and that obesity accelerates methylation drift[23, 61]. Thus, it is a plausible hypothesis that obesity increases the risk of diseases through accelerated epigenetic tissue aging, while CR dampens this risk through an opposite effect.

In conclusion, we find striking conservation of methylation drift with aging among species and the strong negative correlation between methylation drift and lifespan across several species. The CR effects on age-related methylation of delaying the drift may be important to the health and life extension seen in CR animals. Thus, we propose that DNA methylation drift is one of the strongest known biomarkers of lifespan. It is worth investigating whether interventions that further slowdown age-related DNA methylation drift may have beneficial effects on longevity and/or preventing the progression of age-related diseases.

## Methods

**Tissue samples**. We studied a total of 43 mouse (female; $n = 23$, male; $n = 20$) and 57 rhesus monkey (female; $n = 26$, male; $n = 31$) blood samples including animals fed normally (ad libitum; AL) and CR (mouse; AL; $n = 31$, CR; $n = 12$, monkey; AL; $n = 39$, CR; $n = 18$). All mice were purchased from the CR rodent colony at the National Institutes on Aging, National Institutes of Health. All rhesus monkeys were part of a larger longitudinal CR project at the Wisconsin National Primate Research Center. We also examined cord blood or whole blood cells from 139 humans (all healthy individuals; Japanese; $n = 123$, unknown; $n = 16$; sex; female; $n = 57$, male; $n = 69$, unknown; $n = 13$). We selected the size of samples based on our previous studies of age-related methylation in mice[20, 44]. All research with mice was reviewed and approved by the Institutional Animal Care and Use Committee (IACUC) of Temple University. All non-human primate samples were collected at the Wisconsin National Primate Research Center under a protocol approved by the IACUC of the Graduate School of the University of Wisconsin, Madison. All human samples were collected under protocols approved by the institutional review boards of the involved institutions (MD Anderson Cancer Center, Temple University and Fujita Health University School of Medicine) and all subjects provided written informed consent for the collection of residual tissues as per institutional guidelines and in accordance with the Declaration of Helsinki. We summarized sample information including age and caloric status, and assays performed in Supplementary Tables 1, 2. We analyzed female and male samples together to increase statistical power, since there were no statistically significant differences between the sexes in age-related methylation. Human granulocytes were separated by gradient centrifugation to ~98% purity. Polyclonal activated T-cells were obtained from the mononuclear cell fraction and in vitro expanded using Human T-Activator CD3/CD28 Dynabeads (Gibco). All animals and human subjects were clinically healthy (disease-free) at the time of sample collection. We isolated genomic DNA using standard procedures. Briefly, the tissue was digested in a lysis buffer (10 mM TrisHCl, pH 8.0, 10 mM NaCl, 10 mM EDTA, 1% SDS) containing proteinase K (500 µg/mL) overnight at 50 °C, extracted with phenol/chloroform (1:1), and precipitated with 100% ethanol. The resulting pellet was then washed with 70% ethanol, dried, and dissolved in distilled deionized water.

**DREAM**. DREAM was performed as described previously[32, 33]. Briefly, genomic DNA samples were spiked in with methylation standards, sequentially digested with the SmaI (methylation sensitive) and XmaI (methylation insensitive) restriction endonucleases creating methylation-specific signatures at the ends of the

restriction fragments based the CpG methylation status at CCCGGG target sites. After the digestion and ligation of sequencing adaptors, the libraries were sequenced on Illumina Gene Analyzer II or Illumina HiSeq 2000 at the MD Anderson Center for Cancer Epigenetics and HiSeq 2500 at the Fox Chase Cancer Center. Sequencing reads were mapped to the reference genome (mm9, rheMac2, hg19), the reads with unmethylated GGG and methylated CCGGG signatures at individual target CpG sites were counted and the methylation values were adjusted based on spiked in standards. We used for further analyses CpG sites covered with $\geq 100$ reads in at least 75% samples located on autosomal chromosomes. Potential individual SNPs at target CCCGGG sites would not affect the data analysis. Since a single-nucleotide polymorphism will destroy the *SmaI/XmaI* target site, the polymorphic allele would not be included in the analysis. In the case of a homozygous single-nucleotide polymorphism, the CpG site of the affected individual would not be represented. To assess the reproducibility of the method, we performed replicate analyses of identical DNA sample from the normal human whole blood. There was a high concordance among seven technical replicates (Pearson correlation $r \geq 0.997$; Supplementary Fig. 17). The DREAM data are deposited in the GEO database (GSE75499) (http://www.ncbi.nlm.nih.gov/geo/query/acc.cgi? token=itmlcimapdadbsb&acc=GSE75499).

**Permutation analyses**. In order to evaluate the association between age and methylation level, we calculated the Spearman's correlation coefficient ($r$) between age and methylation level of each CpG site measured in the DREAM assay. To assess the statistical significance of these correlations, a permutation approach was used. Briefly, the age was shuffled 1000 times, then the correlation between these randomly shuffled values and methylation level was calculated. Empirical $p$-values were calculated by comparing the observed Spearman's $r$-value against the distribution of $r$-values calculated after random age shuffling, and were equal to the number of permutations with higher correlations than the observed correlations divided by 1000. Next, we used $|r| \geq 0.5$ and empirical $p$-value $< 0.05$ as thresholds to select CpG sites showing age-related drifts. All statistical analyses were performed independently in each species (mouse, monkey, or human) using the R statistical framework (www.r-project.org/).

**Pathway analyses**. Functional class annotation analysis was performed on hyper- or hypo-methylated genes by using the Ingenuity Pathway Analysis software. We analyzed biological processes, molecular functions, and cellular components that were relatively enriched by the gene lists of interest.

**Differential expression analysis by RNA-seq**. RNA-seq data were downloaded from Gene Expression Omnibus (GEO) Series GSE60216. The reads were mapped to the human genome (hg18) by TopHat (version 2.0.10)[62]. The number of reads in each known gene from RefSeq database[63] (downloaded from UCSC Genome Browser on 02 June 2014) was enumerated using htseq-count from HTSeq package (version 0.6.1) (http://www-huber.embl.de/users/anders/HTSeq/). The differential expression between conditions was statistically assessed by R/Bioconductor package DESeq (version 1.16.0)[43]. Genes with $p$-value $< 0.05$ and fold change $>2$ were called significant.

**Bisulfite pyrosequencing for DNA methylation analysis**. Bisulfite treatment of genomic DNA was performed using the EpiTect Bisulfite Kit according to the manufacturer's instructions (Qiagen). We used a quantitative bisulfite pyrosequencing method for DNA methylation analyses as reported previously[20, 44]. In brief, bisulfite-treated DNA was amplified with gene-specific primers in a two-step PCR. The second step of PCR was used to label the reverse DNA strand with biotin. DNA methylation was measured as the percentage of bisulfite-resistant cytosines at CpG sites by pyrosequencing. Pyrosequencing was performed using the PyroMark Gold Q96 CDT Reagents (Qiagen) on the PyroMark Q96 MD platform (Qiagen). Pyro Q-CpG Software (Qiagen) was used to analyze the data. Primer sequences and PCR conditions for bisulfite pyrosequencing assays are listed in Supplementary Table 20. The data points represent averages of bisulfite PCR/pyrosequencing assays performed in duplicates or triplicates.

**Age-related methylation drift**. We used an R package lme4[64], to build a multi-level mixed linear model including data from all 10 hypermethylated ARM genes homologous across all three species, including species as a fixed effect with an interaction term for age allowed to give the difference in methylation rate between genes and species.

To test the CR effect on aging methylation drift, multiple linear regression for methylation with two predictors and their interaction term was performed for each site in DREAM data: age (quantitative variable) and diet (qualitative variable with two levels: AL and CR). Taking AL as the baseline, if the coefficient for the interaction term is significantly non-zero, it indicates CR significantly changes the rate of methylation drift.

**Age prediction**. We derived a linear model for chronological age based on methylation values of 24 hypermethylated genes in AL mice characterized by the equation based on the values of slope and intercept calculated by the multilevel mixed linear model in whole blood (Supplementary Table 17). By substituting the values of the average methylation percentage (24 genes) of individual CR mice into

the equation, we obtained the predicted epigenetic age. We also predicted the ages of CR mice based on methylation status using the linear models calculated using age and methylation status average of 12 hypermethylated ARM genes in AL animals across multiple tissues (coefficient and intercept for age prediction are shown in Supplementary Table 17). Using bisulfite pyrosequencing results of 24 hypermethylated genes, we also made a linear model to predict the age of CR old monkeys based on the values of slope and intercept shown in Supplementary Table 17.

**Quantitative RT-PCR assay for gene expression**. Total RNA from young and old liver samples was prepared by using the RNeasy Mini Kit (Qiagen) and reverse transcribed into complimentary DNA using random hexamer primers and the high-capacity complimentary DNA reverse transcription kit (Applied Biosystems) according to the manufacturer's directions. The expression of genes was quantified using TaqMan gene expression assays and a StepOne Real-Time PCR system (Applied Biosystems). Gene expression was normalized to *Gapdh*. Assay IDs are following (*Elovl2*; Mm00517086_m1, *Ripk4*; Mm00458366_m1, *Lims2*; Mm00523019_m1, *Tapbp*; Mm00493417_m1, *Gapdh*; Mm99999915_g1, Applied Biosystems).

**Measurement of telomere length by quantitative PCR**. We performed quantitative PCR assay to determine the relative telomere length[65–67]. Telomere repeat copy-number/reference gene (*RPLP0*; *36B4*, or *HBG1*) copy-number values were calculated by the formula $2^\wedge -dCt$ where $dCt =$ Ct telomere-Ct reference gene. The DNA samples (10 ng) were assayed in triplicate in 20 µl reaction volume using SYBR Kit (Bio-Rad). The oligonucleotide primers are shown in Supplementary Table 21. The PCR conditions for amplification (using Telomere-1 primers and 36B4 primers) were: 95 °C for 10 min followed by 40 cycles at 95 °C for 15 s, and 60 °C for 1 min. To amplify telomere by Telomere-2 primers, the same conditions were used except annealing temperature as 54 °C. The PCR conditions for *HBG1* gene amplification were: 95 °C for 10 min followed by 40 cycles at 95 °C for 15 s, and 58 °C for 1 min. The quantitative PCR was performed using a StepOne Real-Time PCR system.

**Statistics**. Spearman correlation between methylation and age was calculated using GraphPad Prism 5.0 (GraphPad Software Inc., San Diego, CA, USA). Correlation analysis for DREAM data was performed using the statistical software package R. All $p$-values are two-sided with a $p < 0.05$ considered to be significant. Hierarchical clustering was performed by ArrayTrack Software available at http://edkb.fda.gov/webstart/arraytrack/ using Ward's method. $p$-values for comparisons between sample groups based on age/caloric status in each species were obtained using the unpaired $t$-test with Welch's correction. A $\chi$-test using 2×2 tables was used to calculate $p$-values for the significance of the overlaps.

**Data availability**. The DREAM data discussed in this publication have been deposited in NCBI Gene Expression Omnibus with the accession number GSE75499. All other relevant data are available from the corresponding author upon request.

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

## Acknowledgements

The high-throughput sequencing was supported by Core grant CA016672 (SMF) to The University of Texas MD Anderson Cancer Center. The Genomic Facility at Fox Chase Cancer Center was supported by Core Grant CA006927 to the Fox Chase Cancer Center. This work was supported by National Institutes of Health grants CA158112 (to J.-P.J.I.) and NIH/NIA grants AG11915, and AG040178 (to R.J.C.) and by a grant from the Ellison Medical Foundation (to J.-P.J.I.). Research reported in this publication was supported in part by the Office of the Director, National Institutes of Health under Award Number P51OD011106 to the Wisconsin National Primate Research Center, University of Wisconsin-Madison. This research was conducted in part at a facility constructed with support from Research Facilities Improvement Program grant numbers RR15459-01 and RR020141-01. J.-P.J.I. is an American Cancer Society Clinical Research professor supported by a generous gift from the F. M. Kirby Foundation.

## Author contributions

S.M. conceived, and S.M. and J.-P.J.I. designed and supervised this study. S.M., T.T., and R.J.C. collected tissues. S.M., T.T., and J.T.L. performed experiments. S.M., Y.L., J.M., S.L., and J.J. performed statistical and bioinformatic analyses. S.M. and J.-P.J.I. wrote the manuscript.
