## [Peer Review File · Nature Communications]

Reviewers' comments:

Reviewer #1 (Remarks to the Author):

This is an interesting manuscript that describes observations on the association of ageing, caloric restriction and patterns of DNA methylation. The work is novel and of wide interest.

In review, the major issue with the work is related to the fact that the entirety of the observations and associations presented are cross-sectional. That is, it is not possible to truly show that the changes observed are causally related to age or caloric restriction without actually following the animals/people over time. This is an important limitation that must be acknowledged. Prospective studies are needed.

A puzzling part of the work presented here has to do with the relationship of 'age-related epigenetic drift' with calculated 'methylation age', and methylation as a reflection of cellular mitosis (which is loosely associated with age). Since tissues and tissue stem cells have well known and strikingly different mitotic rates, methylation age cannot reflect the process of mitosis (in fact, the metric for methylation age was developed to be tissue independent, making reflecting mitosis impossible). It is not clear that epigenetic drift is at all similar, yet the data presented here would suggest that the rates of drift correlate with age. While the authors suggest that they may be measuring an effect related to stem cells ("stem cell plasticity") this would require either that the loci they interrogate are stem cell specific (i.e. not confounded by somatic changes) or that any drift that occurs in mitotically active somatic cells is profoundly different. One assumes that they are actually measuring methylation that reflects both changes that are of stem cell origin and methylation that is somatically acquired – how can they be distinguished? This is mechanistically puzzling; clearly epigenetic drift is different in its origin from methylation age. Further complicating this is the work suggesting that stem cell divisions and mutations are the main drivers of cancer. It would be helpful if the authors could add to their discussion these somewhat subtle ideas and address their data and its relationship to stem cell changes as well as non-stem cell changes, attempting to harmonize these somewhat orthogonal ideas.

One additional issue is related to the possibility that tissue variation and diversity is related to caloric restriction (CR). CR related changes in the immune response would alter DNA methylation profile in blood, potentially acting as a confounder of the blood data presented here. Similarly, CR associated differences in histology/pathology (e.g. immune infiltration, tissue or cell-type changes in apoptosis, amount and distribution of brown vs. white fat, efficacy of digestions, constitutive alterations in metabolism in tissues, etc., etc.) may result in potential confounding methylation profiles that would be difficult or impossible to assess and control. As noted below, I am concerned that this potential source of confounding has not been dealt with adequately.

The examination of expression shows a limited, but significant, association of expression with drift, using data from the literature. While interesting, why did the authors not actually use the actual samples to study expression? Direct interrogation would seem preferable.

Assuming that the differences in expression reported are real, do the authors suppose that this reflects a small number of cells with large differences in methylation as opposed to the action of the quite small changes in a large number of cells? Just how would they propose that these changes arise, given that this is supposed to be related to "drift"? One or two loci with differing methylation seems inadequate to change expression patterns as globally as is implied here.

Specific issues:

While the examination of the separated blood cell types alone is of interest, it is not clear how this work was actually done. The number of age-related methylation sites for whole blood differs for each cell subtype comparison; the reason for this is unclear. Further, as there are numerous different subtypes of "T cells", the analysis of these cells is less compelling. It is known that the

aging immune system changes in its character and composition and as a result one would expect this to affect the analysis. Using the limited data presented to assert that there is no confounding and then to ignore potential subtype effects is inadequate.

In suppl Table 7 it would be useful to know how many genes were detected by prior work that were not detected by DREAM analysis in the current manuscript.

Suppl Fig 3 could use some additional explanation of the numbers. For example, the denominators in each comparison appear to be the number of gene promoters with sequence data (I believe). If correct this should be in the legend. Is this all of the promoters with high quality data in each data set? It is difficult to actually understand how the genes are chosen for this analysis.

The pyrosequencing data reporting methylation levels in newborns through elderly people is poorly presented. It is very difficult to actually figure out how many subjects were studied (Suppl Table 1) and there is no description of how these were chosen and where they come from that I can find. The statistical variability in the data used in this analysis is nowhere described (the mean is given but no SD). I also would be concerned about the possibility of additional confounding by cell type in this analysis; for example, early gestational age newborns have significant levels of CD71 positive nRBCs that will have a unique methylation profile which is not controlled for in this analysis. The data shown for the linear extrapolation is not impressive; cell composition should be better accounted for in the analysis.

p-values throughout the suppl data reflect false precision (there is no need to show p-values to 4-5 significant figures) and the type of statistical test used is not stated (e.g. suppl fig 12, 13, 14, 15, 16, 17).

P3 line 47 – a reference might be useful for the assertion that 80% of all CpG sites are methylated.

P3 line 58 – similarly, a reference for epigenetic changes as “initiators” of cancer.

Are there limitations to applications of DREAM across individuals related to SNPs? If so, how is this dealt with? This likely introduces issues in comparison across individuals.

Reviewer #2 (Remarks to the Author):

The authors examine changing patterns of DNA methylation and their association with age from multiple species and tissues as well the effects of caloric restriction on methylation.

1. While the conclusions drawn from their analysis are potentially exciting, I am concerned that the initial survey of methylation sites on which the rest is based is underpowered. The authors should consider a power calculation for the comparison of two binomial samples. The choice of p_0 and p_1 could be based on numbers provided in lines of 84-85 of the manuscript. Selecting appropriate size and power for the tests is somewhat arbitrary but, the size of the test should account for multiple testing and $\alpha=10^{-4}$ would be a minimal standard and $\alpha=10^{-6}$ would be adequate. Whether the authors aim for a higher powered study (power=0.8) or modest power (0.5), I believe that the calculation will speak for itself.

2. The statistical methods applied to these are poorly described in places and misapplied in others. The authors compare methylation drift across species (lines 112-123). Calculation of the overlap between human and monkey sites is incomplete - the authors should construct the full 2x2 table (with human M+/M- on one margin and monkey M+/M-) on the other. It is not clear why the author believe the analysis is "confounded" and not clear how they derived the p-value.

The authors make frequent use of hierarchical clustering to support their conclusions. There are many potential confounders that could cause samples to cluster together. The choice of animals and methylation sites used to address different questions is different for every analysis and it is difficult to evaluate where confounding might be a problem.

The conclusions of the study are of great interest but the underlying logic for different comparisons is not clearly explained and can be confusing to follow.

Reviewer #3 (Remarks to the Author):

Maegawa and colleagues hypothesize that DNA methylation changes with age (methylation drift) are a determinant of lifespan. To test this they collect age, gene expression and DNA methylation data from mice, monkeys and humans. Some of the mice and monkeys had caloric restriction (CR), a known positive factor for lifespan. The authors have collected excellent data but unfortunately their statistical analysis has some major flaws:

Age related methylation correlates with lifespan:

They claim that the rate of methylation drift is associated with lifespan. However, their correlation of -0.99 comes from three data points. The two variables are (i) the log maximum lifespan of a mouse (~5yr), monkey (~70yr) and human (~100yr) (ii) the log of methylation rate in %/year. It is unclear to me where the longevity numbers come from. The latter number was derived by first collapsing 10 measures of methylation into an average, and then regressing methylation on age for each mouse, monkey and human. The authors should employ a random effect model to handle the 10 repeated measures of methylation within subjects, rather than getting the average across genes. More importantly, I would advise the conclusions regarding this association be tempered since the result is based on three observations and is thus speculative. It would be possible to fit one multilevel model including data from all 10 genes across all species, including species as a fixed effect with an interaction term for age allowed to give the difference in methylation rate between species.

CR delays DNA methylation drift:

The authors find CR is related to methylation drift using a two-stage process. For each CpG site, the authors first take the mean difference (across individuals) in methylation between CR and AL animals. They then find the change in methylation per year in AL animals and report the correlation between this and the average difference. It is unclear to me why the x-axis of Figure 3c is AL animals only, and not both. Further, the authors could have tested whether CR was related to methylation drift for each CpG site by a regression of methylation on age with an interaction for CR allowed. These regression results would provide p-values for each CpG site, which could be used to investigate which genes had differential methylation drift by CR group.

The authors develop an age prediction model using data from AL mice/monkeys. They use this model to predict methylation age in CR mice/monkeys. Ignoring the low sample size this is a good idea. Unfortunately, they have set this up in the wrong direction or at least written their model incorrectly. They seem to set methylation as the outcome and age as the predictor variable, but are trying to predict age. For mice they find $\text{methylation} = 0.4 * \text{age} + 13$. From this I am not convinced by the authors' methylation age estimates and subsequent findings of lower methylation age in CR animals.

Age related methylation drift is conserved across species:

The finding that CpG islands where methylation is low in early age tend to see hypermethylation with age, and that CpG islands where methylation is high in early age tend to see hypomethylation

with age could be seen as regression to the mean. This needs to be discussed in more detail. Further, in the p-values for these differences over age (i.e. middle of page 4) come from a test which has not been mentioned in the methods section. Similarly, the tests regarding overlap of hypermethylated genes between species at the top of page 6 are not explained in the methods, but low p-values are found throughout.

Reviewer #4 (Remarks to the Author):

This paper presents epigenetic drift data from mice, monkeys and humans and concludes that it represents a biomarker of lifespan which is attenuated by CR.

In the introduction, "lifestyle" is used twice (first sentence and last paragraph) to characterize CR. I don't think it reflects a lifestyle for most organisms, including the mice and monkeys described in this paper. It would be better to keep the description as an "intervention".

Line 252: The lack of a CR effect on telomere length is described as "Strikingly", but follows with a reference that reported the same lack of an effect. Thus, it doesn't seem so striking.

Line 285: It states that the monkey studies differed with the control diet, my understanding is that it was control and CR diet differences. Thus, the statement here may be an oversimplification.

Reviewers' comments:

Reviewer #1 (Remarks to the Author):

This is an interesting manuscript that describes observations on the association of aging, caloric restriction and patterns of DNA methylation. The work is novel and of wide interest.

-Q1-

In review, the major issue with the work is related to the fact that the entirety of the observations and associations presented are cross-sectional. That is, it is not possible to truly show that the changes observed are causally related to age or caloric restriction without actually following the animals/people over time. This is an important limitation that must be acknowledged. Prospective studies are needed.

-A1-

We agree that causation is difficult to prove and, as suggested by the reviewer, we now acknowledge the limitation of the cross-sectional assays in the discussion section. We note that, as a prospective study to support our cross-sectional data, we have analyzed intra-individual changes of methylation status over time (5 years apart) in monkey peripheral blood mononuclear cells derived from four individuals (Suppl Fig.7) and found results consistent with age-related methylation changes, but the number of samples used is obviously too small to be conclusive.

We modified the text on page 13 as shown below:

“One of the limitations of this study is the reliance on cross-sectional samples. While we did observe methylation changes in a limited number of paired samples obtained from the same individual in the span of five years, longitudinal prospective studies are needed to confirm our observations.”

-Q2-

A puzzling part of the work presented here has to do with the relationship of ‘age-related epigenetic drift’ with calculated ‘methylation age’, and methylation as a reflection of cellular mitosis (which is loosely associated with age). Since tissues and tissue stem cells have well known and strikingly different mitotic rates, methylation age cannot reflect the process of mitosis (in fact, the metric for methylation age was developed to be tissue independent, making reflecting mitosis impossible). It is not clear that epigenetic drift is at all similar, yet the data presented here would suggest that the rates of drift correlate with age. While the authors suggest that they may be measuring an effect related to stem cells (“stem cell plasticity”) this would require either that the loci they interrogate are stem cell specific (i.e. not confounded by somatic changes) or that any drift that occurs in mitotically active somatic cells is profoundly different. One assumes that they are actually measuring

methylation that reflects both changes that are of stem cell origin and methylation that is somatically acquired – how can they be distinguished? This is mechanistically puzzling; clearly, epigenetic drift is different in its origin from methylation age. Further complicating this is the work suggesting that stem cell divisions and mutations are the main drivers of cancer. It would be helpful if the authors could add to their discussion these somewhat subtle ideas and address their data and its relationship to stem cell changes as well as non-stem cell changes, attempting to harmonize these somewhat orthogonal ideas.

-A2-

The reviewer raises key questions in the field and an elaborate discussion of these issues would be more appropriate for a review paper than a paper like this based on space limitations. One fundamental issue is the distinction between “epigenetic (methylation) drift” and “methylation age developed to be tissue independent” (which we assume to be the “methylation clock” derived by Horvath). It is not clear to us that these are fundamentally different and at least one hypothesis is that the “clock” is simply a subset of the sites that show “epigenetic drift”. This subset of sites may be robust because they drift in all tissues or perhaps because they are more cleanly measured by the 450K arrays. In any case, this clock is not easily measurable across species and since we have no data on this, it would be more appropriate to reserve this discussion for another venue.

It is very difficult indeed to relate either drift or the clock to stem cells because tissues examined (by either method) are a mixture of cells (stem/non-stem). The argument that drift is related to adult stem cell turnover is elaborated in review papers (e.g. Issa, JCI 2014; Ref#16) and Ref#53 provides supportive data for this in the hematopoietic system. To clarify our thoughts, the hypothesis is that changes can occur in both differentiated and adult stem cells but that the former changes are rapidly lost due to cell turnover. Therefore, the most reproducible changes that show linear relationships with age likely originate in adult stem cells and are carried into derived somatic/differentiated cells. It is not easy to relate data on tissue specific stem cell turnover to tissue specific epigenetic drift because our understanding of “adult stem cells” is still evolving and the relevant cells are not exactly defined in every tissue. For example, in the large intestine, there are two pools of stem cells, one with high turnover and one with low turnover. Which rate would one use for the relevant correlations? Our data showing that the large intestine is among the highest “drifters” are consistent with the idea that the proliferative stem cells are responsible for day to day activities and the quiescent ones perhaps reserved for injury repair. Similar data can be found in the hematopoietic system. This complex issue is important but again, beyond the scope of the discussion here (in our opinion).

We have added to the discussion in this regards and now state on page 14 as follows:

“While we analyze cells and tissues that are a mixture of stem cells and differentiated cells, it is likely that the most reproducible changes observed are happening in adult stem cells and are carried into differentiated somatic cells¹⁶. Changes that are restricted to

differentiated cells would likely be lost upon cell death and contribute to increasing noise with age but not to the most dramatic and reproducible associations between methylation and age. The “stemness” origin of methylation drift in the hematopoietic system was experimentally established by prior studies⁵³, and differing stem cell turnover rates may explain the tissue specificity we observed – highest drift in large intestine for example (high stem cell turnover).”

References:

(Ref#16); Issa, J. P. Aging and epigenetic drift: a vicious cycle. J. Clin. Invest. 124, 24-29 (2014).

(Ref#53); Beerman, I. et al. Proliferation-dependent alterations of the DNA methylation landscape underlie hematopoietic stem cell aging. Cell Stem Cell 12, 413-425 (2013).

-Q3-

On additional issue is related to the possibility that tissue variation and diversity is related to caloric restriction (CR). CR-related changes in the immune response would alter DNA methylation profile in blood, potentially acting as a confounder of the blood data presented here. Similarly, CR associated differences in histology/pathology (e.g. immune infiltration, tissue or cell-type changes in apoptosis, amount and distribution of brown vs. white fat, efficacy of digestions, constitutive alterations in metabolism in tissues, etc., etc.) may result in potential confounding methylation profiles that would be difficult or impossible to assess and control. As noted below, I am concerned that this potential source of confounding has not been dealt with adequately.

-A3-

We agree that tissue variation and diversity could be a confounder of our results. If this were true, there should be a large overlap between loci that change with age and loci that show cell-type specific methylation. We tested this directly by comparing methylation in granulocytes, CD34+ cells and T-cells vs whole blood cells (WBCs). In these analyses (Suppl Fig.2), only a small degree of overlap was seen between cell type specific changes and age-related methylation changes in WBCs. About 90% of age-related loci in WBCs showed no evidence of tissue-specific methylation in granulocytes, CD34+ cells or T-cells. Moreover, age-related methylation affects both CpG islands (CGIs) and non-CGIs while tissue-specific methylation affects predominantly non-CGI sites (Ref#40). Thus, it is likely that the changes observed here are independent of tissue composition, as has been found by others (Ref#6). However, we acknowledge that it would be useful in future studies to look at this issue directly using purified cell populations.

References:

(Ref#40); Ziller, M. J. et al. Charting a dynamic DNA methylation landscape of the human genome. Nature 500, 477-481 (2013).

(Ref#6); Bell, J. T. et al. Epigenome-wide scans identify differentially methylated regions for age and age-related phenotypes in a healthy ageing population. PLoS Genet. 8, e1002629 (2012).

We modified the text on page 5-6 as follows:

“Moreover, age-related methylation affects both CGIs and non-CGIs while tissue-specific methylation affects predominantly non-CGI sites⁴⁰. Thus, it is likely that the changes observed here are independent of tissue composition.”

We modified the text on page 13 as follows:

“Additionally, DNA methylation studies in pure cell populations would better address potential cell-specific effects on the methylation drift.”

Regarding the histology/pathology, all animals/humans were healthy (diseases-free) as described in the sample section.

We modified the text on page 16 as shown below:

“All animals and human subjects were clinically healthy (disease-free) at the time of sample collection.”

Finally, we used 15 genes showing age-related changes in WBC and observed the effects of CR on their methylation drift in multiple tissue types as well. Therefore, we believe that the issue of tissue/cell specificity as a confounder has limited effects, while the impact of CR is variable depending on tissue type.

We modified the text on page 12 as follows:

“These data also proved that age-related methylation changes occur in multiple tissues and the effect of tissue composition is limited.”

-Q4-

The examination of expression shows a limited, but significant, association of expression with drift, using data from the literature. While interesting, why did the authors not actually use the actual samples to study expression? Direct interrogation would seem preferable.

-A4-

We examined direct expression correlations in a limited number of samples (liver, see Fig. 6) but the study would be considerably more complex if we had analyzed expression in all samples (the paper examines methylation in a total of 427 samples tested). Arguably, finding correlations with publicly available data generated by others increases the validity/generalizability of the findings (Suppl Fig. 3 and 4).

-Q5-

Assuming that the differences in expression reported are real, do the authors suppose that this reflects a small number of cells with large differences in methylation as opposed to the action of the quite small changes in a large number of cells? Just how would they propose that these changes arise, given that this is supposed to be related to “drift”? One or two loci with differing methylation seem inadequate to change expression patterns as globally as is implied here.

-A5-

This is an important point. Our current findings and previous data suggested that drift represents small heterogeneous methylation changes in a large number of cells. We have previously shown by bisulfite PCR cloning/sequencing that methylation of a few CpG sites randomly distributed at multiple alleles correlates with profound effects on gene expression for some genes (Ref#44). We are in the process of analyzing this issue further but we feel this is beyond the scope of the current manuscript (which is already very large).

Reference:

(Ref#44); Maegawa, S. et al. Widespread and tissue specific age-related DNA methylation changes in mice. Genome Res. 20, 332-340 (2010).

-Q6-

Specific issues:

While the examination of the separated blood cell types alone is of interest, it is not clear how this work was actually done. The number of age-related methylation sites for whole blood differs for each cell subtype comparison; the reason for this is unclear. Further, as there are numerous different subtypes of “T cells”, the analysis of these cells is less compelling. It is known that the aging immune system changes in its character and composition and as a result, one would expect this to affect the analysis. Using the limited data presented to assert that there is no confounding and then to ignore potential subtype effects is inadequate.

-A6-

The revised manuscript now includes detailed information about sample collection and actual analyses.

We modified the text on page 16 as follows:

“Human granulocytes were separated by gradient centrifugation to ~98% purity. Polyclonal activated T cells were obtained from the mononuclear cell fraction and in vitro expanded using Human T-Activator CD3/CD28 Dynabeads (Gibco).”

We modified the text on page 5 as follows:

“To assess the effects of blood composition on age-related methylation status detected in whole blood, we performed DREAM methylation analysis using purified subpopulations of blood cells: granulocytes (n=6), CD34+ cells (n=2) and T-cells (n=3), and compared these to whole blood samples (n=16). We detected 222, 1045 and 1923 sites significantly hyper- or hypo-methylated (methylation differences $\geq 2\%$, sequence depth ≥ 100 reads in each site, $FDR < 0.05$) in granulocytes, CD34+ cells and T-cells, respectively, compared to the whole blood. A limited overlap of 0.2%-10% between these cell-type specific sites and the ARM sites determined using whole blood suggested that age-related methylation drift cannot be explained by variability in blood cell subtypes (Supplementary Fig. 2 and Supplementary Table 9).”

We modified the legend of Suppl Fig. 2 as follows:

“Supplementary Figure 2 | A limited effect of blood composition on age-related methylation. Area-proportional Venn diagrams of overlapped CpG sites between sites showing age-related methylation drift in whole blood and sites identified as differentially methylated in blood cell subtypes compared to whole blood. Red number represents the number of sites overlapping. Balloon shows the percentage of overlapped sites of age-related sites in whole blood. To identify the differentially methylated sites between whole blood and each blood cell type, we used sites with sequencing depth ≥ 100 reads in DREAM data among samples. Then, we compared the average of methylation % between the whole blood (n=16; age; 0-86y) and each blood cell type (granulocytes; n=6; age; unknown, CD34+ cells; n=2; age; unknown, T-cells; n=3; age; 19-21y) in each site and defined sites with methylation differences $\geq 2\%$ ($FDR < 0.05$) as differentially methylated sites. A Chi-square test using 2X2 tables (Supplementary Table 9) was used to calculate p-values for the significance of the overlaps. ”

We agree with the comment related to the blood cell subtype composition. It is known that the proportion of cell subtypes changes with age (Ref#A) and that each cell type has a specific methylation status (Ref#B). However, it is also true that age-related methylation drift can be detected using whole blood as shown in numerous publications (Ref #6-9). The CR-induced delay in transition to a more aged-like cell proportion also supports defining age-related methylation drift in whole blood. The concept that age-related methylation correlates with age in whole blood cell samples is still widely accepted. Although we did not compare the age-related methylation among blood cell subtypes, our comparisons support the contention that cell type-specific methylation patterns have limited effects on age-related methylation found in whole blood cells (as shown in Suppl Fig. 2). Moreover, age-related methylation affects both CGIs and non-CGIs while tissue-specific methylation affects predominantly non-CGI sites (Ref#40). Our data, together with the previous reports, prove that there are limited effects on aging to evaluate the aging/CR effects on DNA methylation status by using whole blood cells.

Reference:

(Ref#A); not in the text; Jaffe, A. E. & Irizarry, R. A. Accounting for cellular heterogeneity is critical in epigenome-wide association studies. Genome Biol. 15, R31 (2014).

(Ref#B); not in the text; Reinius, L. E. et al. Differential DNA methylation in purified human blood cells: implications for cell lineage and studies on disease susceptibility. PLoS ONE 7, e41361 (2012).

(Ref#6); Bell, J. T. et al. Epigenome-wide scans identify differentially methylated regions for age and age-related phenotypes in a healthy ageing population. PLoS Genet. 8, e1002629 (2012).

(Ref#7); Horvath, S. et al. Aging effects on DNA methylation modules in human brain and blood tissue. Genome Biol. 13, R97 (2012).

(Ref#8); Hannum, G. et al. Genome-wide methylation profiles reveal quantitative views of human aging rates. Mol. Cell 49, 359-367 (2013).

(Ref#9); Horvath, S. DNA methylation age of human tissues and cell types. Genome Biol. 14, R115 (2013).

(Ref#40); Ziller, M. J. et al. Charting a dynamic DNA methylation landscape of the human genome. Nature 500, 477-481 (2013).

We modified the text on page 5-6 as follows:

“Moreover, age-related methylation affects both CGIs and non-CGIs while tissue-specific methylation affects predominantly non-CGI sites⁴⁰. Thus, it is likely that the changes observed here are independent of tissue composition.”

-Q7-

In suppl Table 7 it would be useful to know how many genes were detected by prior work that were not detected by DREAM analysis in the current manuscript.

-A7-

We have added this information to a new version of the former Suppl Table 7, now Suppl Table 10. We also provided a new supplementary Table 11 showing 2x2 contingency tables of genes overlapping to calculate p-values by Chi-square test.

We added the following columns in Suppl Table 10 and filled them with the number of genes: “Genes hypermethylated by only DREAM” and “Genes hypermethylated in only prior study”.

We provided a new Suppl Table as follows:

“Supplementary Table 11 | Hypermethylated ARM genes detected by DREAM and in prior studies.”

-Q8-

Suppl Fig 3 could use some additional explanation of the numbers. For example, the

denominators in each comparison appear to be the number of gene promoters with sequence data (I believe). If correct this should be in the legend. Is this all of the promoters with high-quality data in each data set? It is difficult to actually understand how the genes are chosen for this analysis.

-A8-

The numbers show the number of promoter genes with high-quality data in each DREAM data set. We counted the number of genes using human homologs of mouse and monkey genes. The denominators represent detectable genes by matching human homologous gene name in each comparison. The revised legend for Suppl Fig.3 (now Fig 2), now includes an explanation of the numbers.

We modified the legend of Fig. 2 as follows:

“Figure 2 | Methylation analysis by DREAM in mouse, monkey, and human DNA. Area-proportional Venn diagrams of overlapping gene promoters (-1kb<TSS<+500bp) showing age-related methylation drift in whole blood in each combination. The denominators represent the number of homologous genes with high-quality sequence data (sequencing depth ≥ 100 reads in 75% of cases) which are detectable between two species in each comparison. We counted the number of genes using human homolog gene names of mouse and monkey genes in each comparison. A Chi-square test using 2X2 tables (Supplementary Table 15) was used to calculate p-values for the significance of the overlaps. p-values are indicated on the right side.”

-Q9-

The pyrosequencing data reporting methylation levels in newborns through elderly people is poorly presented. It is very difficult to actually figure out how many subjects were studied (Suppl Table 1) and there is no description of how these were chosen and where they come from that I can find. The statistical variability in the data used in this analysis is nowhere described (the mean is given but no SD). I also would be concerned about the possibility of additional confounding by cell type in this analysis; for example, early gestational age newborns have significant levels of CD71 positive nRBCs that will have a unique methylation profile which is not controlled for in this analysis. The data shown for the linear extrapolation is not impressive; cell composition should be better accounted for in the analysis.

-A9-

The “number of samples” section in Supplementary Table 1 has been modified. We clarified the gene selection for pyrosequencing assays in Suppl Table 16. We also show a summary of the samples studied in each analysis in Suppl Table 16. We included SEM as an indicator of variability of the data in Suppl Table 17, 18 and 19.

We provided a new Suppl Table as follows:

“Supplementary Table 16 | Samples used in each analysis and gene selection for pyrosequencing assays and gene expression assays.”

We modified the text on page 7 as follows:

“We selected genes based on age-related drift detected in at least one species (Supplementary Table 16) and separately based on prior publications^{20,26-28,44}.”

To answer the reviewer’s specific question, we calculated a Pearson correlation matrix comparing cord blood samples for pyrosequencing data (sample size; n=13, Suppl Table 3). These data suggest a high similarity of methylation status among the samples, although we do not have information on the gestational age of the human cord blood samples. Several research groups have reported age-related methylation drift using cord blood samples as the specimens of earliest time point (Ref#6, 34, 35, 37) and concluded that at least some age-related changes were not a result of compositional changes in the cell types found in whole blood. In addition, a recent study revealed very similar global methylation status between cord blood samples and whole blood samples taken from the same individuals three years later (Ref#36). Moreover, tissue (cell type) specific methylation mainly occurred outside CGIs (Ref#40). Our pyrosequencing assays were performed only for promoter regions at CGI or CpG density high regions to detect the age-related methylation drift. Taken together, this information leads us to believe that cell type composition has limited effects on age-related methylation.

References:

(Ref#6); Bell, J. T. et al. Epigenome-wide scans identify differentially methylated regions for age and age-related phenotypes in a healthy ageing population. PLoS Genet. 8, e1002629 (2012).

(Ref#34); Alisch, R. S. et al. Age-associated DNA methylation in pediatric populations. Genome Res. 22, 623-632 (2012).

(Ref#35); Florath, I., Butterbach, K., Muller, H., Bewerunge-Hudler, M. & Brenner, H. Cross-sectional and longitudinal changes in DNA methylation with age: an epigenome-wide analysis revealing over 60 novel age-associated CpG sites. Hum. Mol. Genet. 23, 1186-1201 (2014).

(Ref#37); Heyn, H. et al. Distinct DNA methylomes of newborns and centenarians. Proc. Natl Acad. Sci. U S A 109, 10522-10527 (2012).

(Ref#36); Herbstman, J. B. et al. Predictors and consequences of global DNA methylation in cord blood and at three years. PLoS ONE 8, e72824 (2013).

(Ref#40); Ziller, M. J. et al. Charting a dynamic DNA methylation landscape of the human genome. Nature 500, 477-481 (2013).

We modified the text on page 4 as follows:

“We used cord blood to represent age zero in humans. Cord blood samples have been used previously for DNA methylation studies in aging^{6,34-37}. DNA methylation profiles in cord blood cell specimens can be potentially affected by the presence of nucleated red blood cells in case of a low gestational age^{38,39}. This was likely not the case in our study because all cord blood samples we tested showed a very high concordance of methylation values (Pearson $r > 0.95$) (Supplementary Table 3).”

References:

(Ref#38); de Goede, O. M. et al. Nucleated red blood cells impact DNA methylation and expression analyses of cord blood hematopoietic cells. Clin. Epigenetics 7, 95 (2015).

(Ref#39); Bohlin, J. et al. Prediction of gestational age based on genome-wide differentially methylated regions. Genome Biol. 17, 207 (2016).

We modified the text on page 5-6 as follows:

“Moreover, age-related methylation affects both CGIs and non-CGIs while tissue-specific methylation affects predominantly non-CGI sites⁴⁰. Thus, it is likely that the changes observed here are independent of tissue composition”

We provided a new Suppl Table as follows:

“Supplementary Table 3 | Pearson correlation matrix between cord blood samples by methylation status detected by 16 pyrosequencing assays.”

-Q10-

p-values throughout the suppl data reflect false precision (there is no need to show p-values to 4-5 significant figures) and the type of statistical test used is not stated (e.g. suppl fig 12, 13, 14, 15, 16, 17).

-A10-

We reduced the number of significant figures shown in the p-values and we described the statistical tests in figure legends and in the text (Results and Methods section).

We modified the text in the “Method” section on page 20 as follows:

“p-values for comparisons between sample groups based on age/caloric status in each species were obtained using the unpaired t-test with Welch’s correction. A Chi-square test using 2X2 tables was used to calculate p-values for the significance of the overlaps.”

-Q11-

P3 line47 – a reference might be useful for the assertion that 80% of all CpG sites are methylated.

-A11-

We added citations (Ref#4) on page 3 (now line 48).

Reference:

(Ref#4); Bird, A. DNA methylation patterns and epigenetic memory. Genes Dev. 16, 6-21 (2002).

-Q12-

P3 line 58 – similarly, a reference for epigenetic changes as “initiators” of cancer.

-A12-

We added citations (Ref#16-18) on page 3 line 58.

References:

(Ref#16); Issa, J. P. Aging and epigenetic drift: a vicious cycle. J. Clin. Invest. 124, 24-29 (2014).

(Ref#17); Li, Y. & Tollefsbol, T. O. Age-related epigenetic drift and phenotypic plasticity loss: implications in prevention of age-related human diseases. Epigenomics 8, 1637-1651 (2016).

(Ref#18); Gonzalo, S. Epigenetic alterations in aging. J. Appl. Physiol. (1985) 109, 586-597 (2010).

-Q13-

Are there limitations to applications of DREAM across individuals related to SNPs? If so, how is this dealt with? This likely introduces issues in comparison across individuals.

-A13-

We modified the text on page 17 as follows:

“Potential individual SNPs at target CCCGGG sites would not affect the data analysis. Since a SNP will destroy the SmaI/XmaI target site, the polymorphic allele would not be included in the analysis. In the case of a homozygous SNP, the CpG site of the affected individual would not be represented.”

Reviewer #2 (Remarks to the Author):

The authors examine changing patterns of DNA methylation and their association with age from multiple species and tissues as well the effects of caloric restriction on methylation.

-Q14-

1. While the conclusions drawn from their analysis are potentially exciting, I am concerned that the initial survey of methylation sites on which the rest is based is underpowered. The authors should consider a power calculation for the comparison of two binomial samples. The choice of p_0 and p_1 could be based on numbers provided in lines of 84-85 of the manuscript. Selecting appropriate size and power for the tests is somewhat arbitrary but, the size of the test should account for multiple testing and $\alpha=10^{-4}$ would be a minimal standard and $\alpha=10^{-6}$ would be adequate. Whether the authors aim for a higher powered study (power=0.8) or modest power (0.5), I believe that the calculation will speak for itself.

-A14-

Following the suggestion of the reviewer, we performed a power calculation using p_1 and p_2 values for methylation ratio of young group and old group, respectively. All values of power were found as 1.0 in all comparisons in each species. We further calculated the sample sizes (coverage of reads per CpG site in each group) using $\alpha=1e-6$ and power=0.8 to get the minimal coverage of reads needed in all cases. Since the numbers of most variable sites used in computing p_1 and p_2 for mouse, monkey, and human were derived from high-quality data (Sequencing depth ≥ 100 reads; independent events), the number of events used in the paper exceed the sample size required in all cases as shown in Suppl Table 4. Finally, a lower power would underestimate the significance of our findings. Thus, we are confident that the results reported for age and CR effects are statistically significant.

We modified the text on page 5 as follows:

“The sample sizes for all comparisons were sufficient to give statistical power >0.8 (Supplementary Table 4).”

We provided a new Suppl Table 4 as follows:

“Supplementary Table 4 | Power and sample size calculations for methylation difference between young and old samples.”

-Q15-

2. The statistical methods applied to these are poorly described in places and misapplied in others. The authors compare methylation drift across species (lines 112-123). Calculation of the overlap between human and monkey sites is incomplete - the authors should construct the full 2x2 table (with human M+/M- on one margin and monkey M+/M-) on the other. It is

not clear why the author believe the analysis is "confounded" and not clear how they derived the p-value.

-A15-

Following the reviewer's suggestions, we described the statistical methods in the text (as follows) and figure legends (for Fig. 2) (as mentioned earlier in the response to reviewer#1 - Q8-). We calculated the significance of overlaps among species (Suppl Table 15) using contingency 2x2 tables and Chi-square test. We deleted the sentence which had "confounded" to avoid confusion.

We provided a new Suppl Table 15 as follows:

"Supplementary Table 15 | Hypermethylated ARM sites overlapping among species."

We modified the legend of Fig. 2 as follows:

"Figure 2 | Methylation analysis by DREAM in mouse, monkey, and human DNA. Area-proportional Venn diagrams of overlapping gene promoters (-1kb<TSS<+500bp) showing age-related methylation drift in whole blood in each combination. The denominators represent the number of homologous genes with high-quality sequence data (sequencing depth ≥ 100 reads in 75% of cases) which are detectable between two species in each comparison. We counted the number of genes using human homolog gene names of mouse and monkey genes in each comparison. A Chi-square test using 2X2 tables (Supplementary Table 15) was used to calculate p-values for the significance of the overlaps. p-values are indicated on the right side."

We modified the text in the "Method" section on page 20 as follows:

"A Chi-square test using 2X2 tables was used to calculate p-values for the significance of the overlaps."

-Q16-

The authors make frequent use of hierarchical clustering to support their conclusions. There are many potential confounders that could cause samples to cluster together. The choice of animals and methylation sites used to address different questions is different for every analysis and it is difficult to evaluate where confounding might be a problem.

-A16-

We agree with these remarks but point out that our conclusions are based on statistical differences (for each age group/caloric status) and age predictions from methylation status, not on clustering assays. We simply used hierarchical clusters to illustrate the trends of methylation status by age and the attenuation of aging by caloric restriction.

We agree that there are potential confounders and discuss earlier the issue of tissue composition. The problem of potential confounders is exactly why we examined multiple

species and multiple tissues – all leading to the same conclusions. To study all confounders in this data set is not possible and in fact to design a study that would account for all possible confounders is not feasible as it would require the use of completely matched animals (e.g. an exceedingly large number of cloned animals) housed and managed in an identical fashion. It is even more difficult to study all confounders in humans because of the huge heterogeneity.

In response to the choice of animals and methylation sites used, we clarified and summarized the sample sizes and genes tested in Suppl Table 16. We began with 10 homologous genes to study the association between lifespan and methylation rate among species by pyrosequencing. To increase the accuracy of our age prediction we then increased the number of genes to 24 (14 in humans). To study the tissue specificity of CR effects in mice, we used 12 genes that showed age-related hypermethylation in mouse whole blood. This is not all 24 genes tested in mouse whole blood, however, we also provided clustering of whole blood using the set of 12 genes (Suppl Fig. 14). We believe that this is a reasonable approach to evaluating tissue specificity.

We provided a new Suppl Table as follows:

“Supplementary Table 16 | Samples used in each analysis and gene selection for pyrosequencing assays and gene expression assays.”

-Q17-

The conclusions of the study are of great interest but the underlying logic for different comparisons is not clearly explained and can be confusing to follow.

-A17-

In the revised manuscript we have improved our explanations (in Suppl Table 1, 2 and newly provided Suppl Table 16 as shown in responses of -A9- and- A16-) for different comparisons.

We provided a new Suppl Table as follows:

“Supplementary Table 16 | Samples used in each analysis and gene selection for pyrosequencing assays and gene expression assays.”

Reviewer #3 (Remarks to the Author):

-Q18-

Maegawa and colleagues hypothesize that DNA methylation changes with age (methylation drift) are a determinant of lifespan. To test this they collect age, gene expression and DNA methylation data from mice, monkeys, and humans. Some of the mice and monkeys had a caloric restriction (CR), a known positive factor for lifespan. The authors have collected excellent data but unfortunately, their statistical analysis has some major flaws:

-A18-

We have carefully revised the statistical analyses in the current version of our manuscript.

-Q19-

Age-related methylation correlates with lifespan:

They claim that the rate of methylation drift is associated with lifespan. However, their correlation of -0.99 comes from three data points. The two variables are (i) the log maximum lifespan of a mouse (~5yr), monkey (~70yr) and human (~100yr) (ii) the log of methylation rate in %/year. It is unclear to me where the longevity numbers come from. The latter number was derived by first collapsing 10 measures of methylation into an average, and then regressing methylation on age for each mouse, monkey, and human. The authors should employ a random effect model to handle the 10 repeated measures of methylation within subjects, rather than getting the average across genes. More importantly, I would advise the conclusions regarding this association be tempered since the result is based on three observations and is thus speculative. It would be possible to fit one multilevel model including data from all 10 genes across all species, including species as a fixed effect with an interaction term for age allowed to give the difference in methylation rate between species.

-A19-

We thank the reviewer for this helpful suggestion. We have recalculated methylation drift using an R package lme4 (Ref#64) to build a multilevel model including data from all 10 genes across all species, including species as a fixed effect with an interaction term for age allowed to give the difference in methylation rate between genes and species. The results are shown in Fig. 3 and Suppl Table 20. We have calculated a multilevel linear mixed-effects model with genes and species as group effects. Using The Animal Ageing and Longevity Database, we restated the maximum longevity of the three species (mouse, 4 years; monkey, 40 years; human, 122.5 years) (Ref#29-31). The source for these ages is now described in the text (as follows). The overall results and conclusions remain unchanged (and some of the associations show even greater significance using this analysis).

We modified the text on page 8 as follows:

“We employed a multilevel linear mixed effect model to calculate methylation drift at 10 homologous genes and obtained slopes representing methylation drift per year in each species (Supplementary Table 20). The drift rates (mean±SEM) were 4.1±1.2%/year in mice, 0.34±0.14%/year in monkeys and 0.10±0.02%/year in humans (Fig. 3a). Methylation drift was thus inversely proportional to longevity (Fig. 3b). ”

We modified the text on page 18 as follows:

“Age-related methylation drift.

We used an R package lme4⁶⁴, to build a multilevel mixed linear model including data from all 10 hypermethylated ARM genes homologous across all three species, including species as a fixed effect with an interaction term for age allowed to give the difference in methylation rate between genes and species.”

Reference:

(Ref#64); Bates, D., Machler, M., Bolker, B. M. & Walker, S. C. *Fitting Linear Mixed-Effects Models Using lme4*. *J Stat Softw* 67, 1-48 (2015).

We modified the text on page 4 as follows:

“The maximum longevity of mice, rhesus monkeys and humans is 4, 40 and 122.5 years, respectively (The Animal Ageing and Longevity Database)²⁹⁻³¹”

References:

(Ref#29); Miller, R. A., Harper, J. M., Dysko, R. C., Durkee, S. J. & Austad, S. N. *Longer life spans and delayed maturation in wild-derived mice*. *Exp. Biol. Med. (Maywood)* 227, 500-508 (2002).

(Ref#30); Mattison, J. A. et al. *Impact of caloric restriction on health and survival in rhesus monkeys from the NIA study*. *Nature* 489, 318-321 (2012).

(Ref#31); Allard, M., Lèbre, V., Robine, J.-M. & Calment, J. *Jeanne Calment : from Van Gogh's time to ours, 122 extraordinary years*. (W.H. Freeman, 1998).

-Q20-

CR delays DNA methylation drift:

The authors find CR is related to methylation drift using a two-stage process. For each CpG site, the authors first take the mean difference (across individuals) in methylation between CR and AL animals. They then find the change in methylation per year in AL animals and report the correlation between this and the average difference. It is unclear to me why the x-axis of Figure 3c is AL animals only, and not both. Further, the authors could have tested whether CR was related to methylation drift for each CpG site by a regression of methylation on age with an interaction for CR allowed. These regression results would

provide p-values for each CpG site, which could be used to investigate which genes had differential methylation drift by CR group.

-A20-

To reveal the effects of CR on age-related DNA methylation status, we purposefully used only AL animals on the x-axis (Figure 3c, now Figure 4c). Our reason for this is now explained in the text.

We modified the text at page 9 as follows:

“To reveal the CR effect, we compared methylation differences in CR versus AL animals to the rate of methylation drift with age in AL mice and found a strong negative correlation (Spearman $r=-0.87$, $p<0.001$, two-tailed, Fig. 4c)”

The reviewer makes an excellent point regarding testing the effects of CR on methylation drift. In response, we have now calculated p-values for each CpG site and investigated which genes show a CR effect on methylation drift. The list of these genes is provided in Suppl Table 21. We included this gene information in Supplementary Table 12 (for mice) and 13 (for monkeys) and explained in the text as follows.

We provided the Suppl Table 21 as follows:

“Supplementary Table 21 | The list of genes for which the methylation rate by age is significantly affected by CR.”

We modified the legend of Suppl Table 12 and 13 as follows:

*“*genes overlapping with genes listed in Supplementary Table 21.”*

We modified the text on page 9-10 as follows:

“We investigated whether CR was related to methylation drift by a multiple linear regression of methylation on age with an interaction for CR allowed. Based on the p-values <0.05 provided by the regression results for each CpG site, we defined genes where age-related drift was significantly alleviated by CR (Supplementary Table 21). As expected, almost every gene detected by this model (Supplementary Table 21) was also detected as undergoing age-related methylation as listed in Supplementary Tables 12 and 13. Most of the genes that showed a significant effect of CR as indicated by negative coefficients overlapped with hypermethylated ARM genes and vice versa. (Supplementary Fig. 11 and Supplementary Table 21). These data suggest that CR may diminish or eliminate methylation changes with age.”

We provided a new Suppl Figure 11 as follows:

“Supplementary Figure 11 | Overlap between genes affected by CR and genes showing age-related methylation.”

We modified the text on page 18 in method section as follows:

“To test the CR effect on aging methylation drift, multiple linear regression for methylation with two predictors and their interaction term was performed for each site in DREAM data: age (quantitative variable) and diet (qualitative variable with two levels: AL and CR). Taking AL as the baseline, if the coefficient for the interaction term is significantly non-zero, it indicates CR significantly changes the rate of methylation drift.”

-Q21-

The authors develop an age prediction model using data from AL mice/monkeys. They use this model to predict methylation age in CR mice/monkeys. Ignoring the low sample size this is a good idea. Unfortunately, they have set this up in the wrong direction or at least written their model incorrectly. They seem to set methylation as the outcome and age as the predictor variable but are trying to predict age. For mice they find $\text{methylation} = 0.4 * \text{age} + 13$. From this, I am not convinced by the authors' methylation age estimates and subsequent findings of lower methylation age in CR animals.

-A21-

We have deleted the equations from the text to avoid confusion and we have now explained how to calculate the predicted age in the legend of Suppl Table 20.

We modified the legend of Suppl Table 20 as follows:

Formulas for age prediction;

$\text{Age}(\text{year}) = (1/\text{Slope}) \times \text{Methylation}(\%) + (\text{Intercept}/\text{Slope})$

-Q22-

Age-related methylation drift is conserved across species:

The finding that CpG islands where methylation is low in early age tend to see hypermethylation with age, and that CpG islands where methylation is high in early age tend to see hypomethylation with age could be seen as a regression to the mean. This needs to be discussed in more detail. Further, in the p-values for these differences over age (i.e. middle of page 4) come from a test which has not been mentioned in the methods section. Similarly, the tests regarding the overlap of hypermethylated genes between species at the top of page 6 are not explained in the methods, but low p-values are found throughout.

-A22-

Regression to the mean would imply a certain degree of order and dynamism to the process. However, methylation drift can be seen as an erosion of highly organized methylation patterns: focal unmethylated/methylated status of CpG islands, distinctly different from the global methylated status of CpG sites in CpG poor regions. Focal hypermethylation/hypomethylation (and global hypomethylation) occurring in aging and

accentuated/accelerated in cancer is a regression to the mean only superficially. Indeed, we previously reported age-related methylation changes (increased and decreased patterns in promoter regions) as increased epigenetic noise (with increased variabilities in older populations) in multiple tissue types in mice (Ref#20, 44). The discussion section of the revised manuscript now includes more detailed description of the methylation changes as follows:

We modified the text on page 8 as follows:

“Hypermethylation and hypomethylation occurring with aging could be seen as a regression to the mean. We previously reported age-related methylation changes (increased and decreased patterns in promoter regions) showing increased epigenetic noise (with increased variabilities in older populations) in multiple tissue types in mice^{20,44}.”

We modified the text on page 13 as follows:

“Changes in DNA methylation occur during the aging process in mammals, and this age-related change in DNA methylation is accelerated in tumorigenesis. Methylation drift can be seen as an erosion of highly organized methylation patterns: focal unmethylated status of CGIs, distinctly different from the global methylated status of CpG sites in CpG poor regions⁴⁰. Consistent with our previous studies^{20,44}, we detected age-related methylation drift in multiple tissue types.”

References:

*(Ref#20); Maegawa, S. et al. Age-related epigenetic drift in the pathogenesis of MDS and AML. *Genome Res.* 24, 580-591 (2014).*

*(Ref#44); Maegawa, S. et al. Widespread and tissue specific age-related DNA methylation changes in mice. *Genome Res.* 20, 332-340 (2010).*

*(Ref#40); Ziller, M. J. et al. Charting a dynamic DNA methylation landscape of the human genome. *Nature* 500, 477-481 (2013).*

As explained in the response to -Q15- (question #2 from reviewer #2), we described how we calculated p-values for hypermethylated genes overlapping among species and provided 2X2 tables. We described how p-values were calculated in the Methods section as follows (in “Statistics” section):

We modified the text on page 20 in method section as follows:

“p-values for comparisons between sample groups based on age/caloric status in each species were obtained using the unpaired t-test with Welch’s correction. A Chi-square test using 2X2 tables was used to calculate p-values for the significance of the overlaps.”

Reviewer #4 (Remarks to the Author):

This paper presents epigenetic drift data from mice, monkeys, and humans and concludes that it represents a biomarker of lifespan which is attenuated by CR.

-Q23-

In the introduction, “lifestyle” is used twice (first sentence and last paragraph) to characterize CR. I don’t think it reflects a lifestyle for most organisms, including the mice and monkeys described in this paper. It would be better to keep the description as an “intervention”.

-A23-

We have deleted “lifestyle” on page 2 and replaced “lifestyle” with “intervention” on page 3.

-Q24-

Line 252: The lack of a CR effect on telomere length is described as “Strikingly”, but follows with a reference that reported the same lack of an effect. Thus, it doesn’t seem so striking.

-A24-

We deleted the word “strikingly” on page 13.

-Q25-

Line 285: It states that the monkey studies differed with the control diet, my understanding is that it was control and CR diet differences. Thus, the statement here may be an oversimplification.

-A25-

This was corrected in the text as follows.

We modified the text on page 15 (line 327) as follows:

“The two studies differed substantially in the diets they used,”

REVIEWERS' COMMENTS:

Reviewer #1 (Remarks to the Author):

I find the author's resubmission quite responsive to prior review.

Reviewer #3 (Remarks to the Author):

The authors have answered all of my concerns and I would advise the paper be accepted for publication.

REVIEWERS' COMMENTS:

Reviewer #1 (Remarks to the Author):

I find the author's resubmission quite responsive to prior review.

Reviewer #3 (Remarks to the Author):

The authors have answered all of my concerns and I would advise the paper be accepted for publication.

Thank you so much for positive comments from Reviewer #1 and Reviewer #3.